# BiCompFL: Stochastic Federated Learning with Bi-Directional Compression

## Abstract

Communication is a prominent bottleneck in federated learning (FL). State-of-the-art accuracy performance under limited uplink communications from the clients to the federator is achieved by stochastic FL approaches. It has been recently shown that leveraging side information in the form of a prior distribution at the federator can drastically reduce the uplink communication cost in stochastic FL. Here, the latest global model distribution serves as a natural prior since it can be shared with the clients under ideal downlink communication from the federator to the clients. Nevertheless, downlink communication is often limited in practical settings, and bi-directional compression must be considered to reduce the overall communication cost. The extension of existing stochastic FL solutions to bi-directional compression is non-trivial due to the lack of a globally shared common prior distribution at each iteration. In this paper, we propose `BICompFL`, which employs importance sampling to send samples from the updated local models in the uplink, and the aggregated global model in the downlink by carefully choosing common prior distributions as side-information. We theoretically study the communication cost by a new analysis of importance sampling that refines known results, and exposes the interplay between uplink and downlink communication costs. We also show through numerical experiments that BICompFL enables multi-fold savings in communication cost compared to the state-of-the-art.

## 1 Introduction

Federated learning (FL) is a widely used and well-studied machine learning (ML) framework, where multiple clients orchestrated by a federator collaborate to train an ML model (McMahan et al., 2017). Communication efficiency, privacy, security, and data heterogeneity are critical challenges in FL that have been extensively studied (Zhang et al., 2021; Wen et al., 2023). FL is a bi-directional process, and with the increasing size of ML models, massive amounts of data are communicated between the federator and the clients. Reducing uplink communication from clients to the federator has been the focus of many studies, mainly within the framework of lossy gradient compression, e.g., (Seide et al., 2014; Alistarh et al., 2017; Isik et al., 2024). However, reducing the cost of *downlink* transmission to communicate the model updates from the federator to the clients has received relatively less attention, even though it is costly and can be a major bottleneck when training over a wireless network. An ongoing body of research aims to study the communication bottleneck in downlink transmission, by combining tools from gradient compression, momentum, and error-feedback, cf., (Stich et al., 2018; Tang et al., 2019; Xie et al., 2020; Amiri et al., 2020; Philippenko & Dieuleveut, 2020; Gruntkowska et al., 2023; Tyurin & Richtárik, 2023; Dorfman et al., 2023; Gruntkowska et al., 2024). However, these works are focused on non-stochastic (or non-Bayesian) settings, whereas the state-of-the-art performance in limited uplink communication scenarios is achieved by stochastic compression methods, such as QSGD Alistarh et al. (2017), QLSD Vono et al. (2022), dithered quantization Abdi & Fekri (2019) or FedPM Isik et al. (2023), in which the clients send samples from a local distribution, and the federator estimates the mean of clients' distributions by averaging these samples. In this work, we study the performance of these stochastic FL frameworks with limited communication in both directions, and obtain state-of-the-art results.

Standard analysis of such uni- or bidirectional compression schemes involves studying the convergence rates of the respective optimization task for a given communication budget. In general, the trade-off between communication cost (or compression cost) and distortion has been extensively

studied in information theory under the framework of rate-distortion theory (Cover & Thomas, 2006). However, in stochastic gradient-based optimization, it is difficult to analyze the effect of finite resolution on the convergence speed. Classical rate-distortion results are not suitable for this analysis, since they rely on the joint compression of many samples and additive distortion measures. As a result, it is also difficult to study the fundamental trade-off between the communication load and the convergence speed.

An alternative stochastic FL approach was proposed in Isik et al. (2024), which applies to a variety of stochastic and Bayesian FL solutions, but also covers the classical stochastic compression mechanisms for standard gradient-based methods. Communication reduction in Isik et al. (2024) is achieved by importance sampling, which rather than sampling locally and using lossy compression to send these samples, directly enables the federator to sample from the updated local distribution. This enables a direct evaluation of the communication cost when a shared common prior distribution, called *side information*, and sufficient common randomness is available between the federator and the clients. When the downlink communication is assumed unlimited, the global model distribution at the federator can be shared with all the clients, and serve as a natural side information, i.e., common prior. However, this is not possible under downlink communication constraints, leading to new algorithmic and analysis challenges, which we address here.

Specifically, in this work, we develop and analyze stochastic FL algorithms with bi-directional compression with FedPM as a notable example. The idea in FedPM Isik et al. (2023) is to collaboratively train a probabilistic mask that determines which weights to maintain from a randomly initialized network. The motivation stems from the *lottery-ticket hypothesis* (Frankle & Carbin, 2019), which claims that randomly initialized networks contain sub-networks capable of reaching accuracy comparable to that of the full network. We utilize importance sampling with appropriate priors, and accurately characterize the uplink and downlink communication costs and the estimation error. We also explore various aspects that affect the performance, such as clients' data heterogeneity, and the existence of shared randomness among all clients. The basic question we address is: *Can joint uplink and downlink compression reduce communication bottlenecks in stochastic FL?* We answer this question in the affirmative, and concretely, our contributions are summarized as follows:

- We propose two algorithms for bi-directional stochastic FL, depending on the availability of shared randomness. The first algorithm is for the case when globally shared randomness is available, and the second is for the case when only private shared randomness between each client and the federator exists. Both algorithms leverage carefully chosen side information to transmit samples from the desired distribution through importance sampling.

- We experimentally validate our algorithms and show order-wise improvements in the communication cost without degradation of the accuracy compared to many baselines. We investigate the role of shared randomness and the choice of side information.

- We provide a novel dedicated theoretical analysis of importance sampling to quantify the communication cost of stochastic FL with bi-directional compression. Our findings go beyond the established analysis of Chatterjee & Diaconis (2018), with refinements for the particular case of Bernoulli distributions (which can be of independent interest). Our proofs allow a practically relevant analysis and provide techniques useful for other distributions.

## 2    PRELIMINARIES: STOCHASTIC FL WITH BI-DIRECTIONAL COMPRESSION

Our proposed algorithm is a stochastic FL algorithm based on FedPM and importance sampling. In what follows, we shortly review these concepts.

**Stochastic Federated Learning**    A set of $n$ clients collaboratively and iteratively train a model, e.g., a neural network, under the orchestration of a federator. Each client $i \in [n]$ possesses a dataset $\mathcal{D}_i$, where we define $[n] := \{1, \ldots, n\}$ for an integer $n$ . We differentiate between homogeneous data, where $\mathcal{D}_i$ is drawn independently from the same distribution for all clients (i.i.d.), and heterogeneous data, where each $\mathcal{D}_i$ may come from a different distribution (non i.i.d.). At each iteration $t$ of the training process, the global model at the federator is described by a probability distribution $\theta_t$ with dimension $d$. After downlink transmission, each client $i$ has an estimate $\hat{\theta}_{i,t}$ of $\theta_t$, and locally optimizes $\hat{\theta}_{i,t}$, e.g., according to a loss function $F(\hat{\theta}_{i,t}, \mathcal{D}_i)$ by (stochastic) gradient descent, to ob-

tain a local model $q_i^t$, referred to as the *posterior*. Compressed versions of the clients' posteriors are transmitted back to the federator in the uplink, where the received posteriors are aggregated using an aggregation rule $R(\cdot)$ to obtain a refined global probability distribution $\theta_{t+1} = R\left(\{\hat{q}_i^t\}_{i\in[n]}\right)$. A simple aggregation rule $R(\cdot)$ is the average over all clients' models. This process is repeated until a certain convergence criterion is met. In many stochastic FL settings, like QSGD or FedPM (Isik et al., 2024), the transmitted client updates $\hat{q}_i^t$ are samples from the posterior distribution $q_i^t$.

**FedPM**  As said, we adapt in this work the federated probabilistic mask training (FedPM) framework introduced by Isik et al. (2023), in which the weights $w$ of the network are randomly initialized at the start of training, and remain fixed. The federator and clients only train a mask, which determines for each parameter whether it is activated or not, i.e., identifying an efficient subnetwork within the given fixed network. The probabilistic masks $\theta_t$ are described by Bernoulli distributions, i.e., $\theta_t \in [0,1]^d$ contains a Bernoulli parameter to be trained for each weight of the network. These parameters determine the probability of retaining the corresponding weights. During inference, the weights $w$ are masked with samples $x^t \in \{0,1\}^d \sim \theta_t$ from the distribution $\theta_t$, i.e., the inference is conducted on a network with weights $w \odot x^t$. In FedPM, clients sample from their locally trained models, and send these samples to the federator, which, in turn, updates the global model by averaging these samples. The communication cost of this scheme remains the same for all iterations, even though the communication cost can be reduced since the KL-divergence between the global model and the locally trained models diminishes as the training progresses.

**Importance Sampling**  Isik et al. (2024) proposed a method, called KL minimization with side information (KLMS), to reduce the cost of transmitting the local models $q_i^t$ to the federator. Consequently, the communication cost depends on the KL-divergence between the desired distribution and the common prior. This method utilizes the common side information available at both the clients and the federator, as well as shared randomness. The idea is that instead of sampling locally and sending the samples to the federator, the federator in the KLMS method samples from the desired distribution through importance sampling. In a nutshell, importance sampling (Srinivasan, 2002) makes use of a common prior to sample from a desired distribution. Consider two distributions $P$ and $Q$, where $P$ is known to both parties, and $Q$ is only known to the client. To make the federator sample from $Q$, both parties sample $n_{\mathrm{IS}}$ samples $\{X_i\}_{i\in[n_{\mathrm{IS}}]}$ from $P$. The client forms an auxiliary distribution $W(i) = \frac{Q(X_i)/P(X_i)}{\sum_{i=1}^{n_{\mathrm{IS}}} Q(X_i)/P(X_i)}$ capturing the importance of the samples. A sample from $W$ is fully described by its index $i$, which can be transmitted with $\log_2(n_{\mathrm{IS}})$ bits, and approximates a sample from $Q$. Chatterjee & Diaconis (2018) shown that importance sampling with posterior $Q$ and prior $P$ requires $n_{\mathrm{IS}}$ to be in the order of $\Theta(\exp(\mathrm{D_{KL}}(Q\|P)))$, where $\mathrm{D_{KL}}(Q\|P)$ denotes the KL-divergence between distributions $Q$ and $P$. In what follows, we will also denote the KL-divergence between two Bernoulli distributions $Q$ and $P$ with parameters $q$ and $p$ by $\mathrm{d_{KL}}(q\|p)$.

## 3    BiCompFL: Communication-Efficient Federated Learning

In this section, we describe our proposed scheme, termed BiCompFL. This scheme is a bi-directional stochastic compression strategy, which leverages side information using importance sampling to reduce both the uplink and downlink communication costs. The scheme relies on the availability of shared randomness between each of the clients and the federator, which can be obtained by a pseudo-random sequence generated from a common seed, shared by the federator and each client. However, we will distinguish between private shared randomness (between a specific client and the federator) and global shared common randomness, between all parties, which is more challenging to have. We assume that all the clients and the federator share the same global model $\hat{\theta}_0$ at initialization. This does not incur any communication cost in the case of globally shared randomness, but requires an initial round of transmitting the model from federator to clients in the absence thereof.

**BiCompFL: The General Algorithm**  Next, we will focus our description on probabilistic mask training, similar to FedPM, where the models of dimension $d$ are described by Bernoulli parameters. However, our method serves as a general framework for other stochastic optimization procedures as well. We start with a general description of our algorithm, which is valid for both cases of global and private shared randomness. In iteration $t = 0$, the clients $i \in [n]$ share with the federator the same global model, i.e., $\hat{\theta}_{i,0} = \theta_0$, for all $i \in [n]$. At iteration $t$, each client $i$ locally trains the model $\hat{\theta}_{i,t}$ in

---

**Algorithm 1** Local Training at Client $i$

---

**Require:** Model $\hat{\theta}_{i,t}$

  1: Map model to scores in the dual space: $\mathbf{s}_{i,t}^{(0)} = \sigma^{-1}(\hat{\theta}_{i,t}) = \log\left(\frac{\hat{\theta}_{i,t}}{1-\hat{\theta}_{i,t}}\right)$

  2: **for** Local iterations $\ell \in [L]$ **do**

  3:     $\mathbf{s}_{i,t}^{(\ell)} = \mathbf{s}_{i,t}^{(0)} - \nabla_{\mathbf{s}_{i,t}^{(\ell-1)}} F(\hat{\theta}_{i,t}^{(\ell-1)}, \mathcal{D}_i)$, where $\hat{\theta}_{i,t}^{(\ell-1)} = \sigma(\mathbf{s}_{i,t}^{(\ell-1)})$

  4: **end for**

  5: Map back to primal space: $q_i^t = \sigma(\mathbf{s}_{i,t}^{(L)})$

---

$L$ local iterations. To enable gradient descent, the model $\hat{\theta}_{i,t}$ is mapped to scores $\mathbf{s}_{i,t}^{(0)}$ in a dual space by the inverse Sigmoid function $\mathbf{s}_{i,t}^{(0)} = \sigma^{-1}(\hat{\theta}_{i,t}) = \log(\hat{\theta}_{i,t}) - \log(1 - \hat{\theta}_{i,t})$. The scores are then trained for $L$ local iterations $\ell \in [L]$ by computing the gradient $\nabla_{\mathbf{s}_{i,t}^{(\ell-1)}} F(\hat{\theta}_{i,t}^{(\ell-1)}, \mathcal{D}_i)$, where the straight-through estimator is used to compute the gradient of the non-differentiable Bernoulli sampling operation based on the distribution $\hat{\theta}_{i,t}^{(\ell-1)} = \sigma(\mathbf{s}_{i,t}^{(\ell-1)})$, i.e., the gradient equals the Bernoulli parameter. By mapping the model back to the primal space, each client $i$ obtains a model update in terms of a posterior $q_i^t = \sigma(\mathbf{s}_{i,t}^{(L)})$. This process is a special instance of mirror descent, which, in the special case of optimizing over Bernoulli distributions, employs point-wise optimization with respect to a KL-proximity term (cf. Appendix B for a short discussion). This directly impacts the communication cost. The client training process is summarized in Algorithm 1.

To convey the model update $q_i^t$ to the federator, each client employs importance sampling in $B$ blocks of size $d/B$ each[1] with a prior distribution $p_{i,u}^t$, which is set to $p_{i,u}^0 = \hat{\theta}_{i,0}$ at iteration $t = 0$ (the choice of $p_{i,u}^t$ for $t > 0$ will be clarified later). For each block $b \in [d/B]$, client $i$ conveys $n_{\mathrm{UL}}$ samples $\{y_{i,\ell}^t\}_{\ell \in [n_{\mathrm{UL}}]}$ of $q_i^t$ to the federator by transmitting for each block $b$ an index $I_{i,\ell}^b$ with $\log_2(n_{\mathrm{IS}})$ bits, where $n_{\mathrm{IS}}$ is the number of samples per block, generated from the prior distribution $p_{i,u}^t$ at both the client and the federator using the available shared randomness. The client selects one of these samples via importance sampling, and its index $I_{i,\ell}^b$ is transmitted to the federator, which can then reconstruct the exact sample $y_{i,\ell}^t$. The samples of all blocks are concatenated for each mask $\ell$. Hence, the federator obtains an estimate of client $i$'s posterior distribution using the empirical average $\hat{q}_i^t = \frac{1}{n_{\mathrm{UL}}} \sum_{\ell=1}^{n_{\mathrm{UL}}} y_{i,\ell}^t$.

By averaging the estimates $\hat{q}_i^t$ for all the clients' models, the federator updates the global model as $\theta_{t+1} = \frac{1}{n} \sum_{i=1}^{n} \hat{q}_i^t$. To transmit the new model to each client $i$, we assume the existence of a common prior $p_{i,d}^t$ shared by the federator and the client. With $p_{i,d}^t$, the federator performs importance sampling in $B$ blocks of size $d/B$ to make client $i$ sample from, and thereby estimate, the latest global model $\theta_{t+1}$. The client samples $n_{\mathrm{DL}}$ masks $\{x_{i,\ell}^t\}_{\ell \in [n_{\mathrm{DL}}]}$, each incurring a communication cost of $B \log_2(n_{\mathrm{IS}})$ bits. An estimate of the updated global model is obtained by concatenating the reconstructed samples for all the blocks $b \in [B]$, and averaging over all masks $\hat{\theta}_{i,t+1} = \frac{1}{n_{\mathrm{DL}}} \sum_{\ell=1}^{n_{\mathrm{DL}}} x_{i,\ell}^t$.

Since the number of clients is typically large, it often suffices to choose $n_{\mathrm{UL}} = 1$. The clients' contributions are averaged at the federator, effectively reducing the noise due to the importance sampling step. This allowed Isik et al. (2024) to theoretically analyze the uplink communication cost for importance sampling-based stochastic communication of model updates. In principle, we will follow a similar approach for downlink communication; however, the main challenge is that downlink communication cannot benefit from the averaging effect of multiple clients, and so we reduce the variance of the model estimate in the downlink by setting $n_{\mathrm{DL}} = n \cdot n_{\mathrm{UL}}$.

The choice of the priors $p_{i,u}^t$ and $p_{i,d}^t$ for importance sampling in the uplink and downlink channels, respectively, crucially affects the performance and the communication cost of the algorithm. As a first-order characterization, the communication cost of importance sampling is determined by $\mathrm{D}_{\mathrm{KL}}(q_i^t \| p_{i,u}^t)$ in the uplink and by $\mathrm{D}_{\mathrm{KL}}(\theta_{t+1} \| p_{i,d}^t)$ in the downlink.

---

[1]Assuming for simplicity that $B|d$.

---

**Algorithm 2** BICOMPFL-GR with Global Shared Randomness

---

**Require:** Both clients and federator initialize the same global model $\mathbf{w}^{(0)}$ using a shared seed
**Ensure:** Clients set prior $\forall i \in [n] : p_{i,u}^t = p_{i,d}^t = \hat{\theta}_0$
  1: **repeat**
  2:    **for** Clients $i \in [n]$ **do**
  3:       Local training of $q_i^t$ according to Algorithm 1
  4:       Client samples indices $I_{i,\ell}^b, \ell \in [n_{\mathrm{UL}}], b \in [B]$ from $q_i^t$ with prior $p^t = \hat{\theta}_t$
  5:    **end for**
  6:    Federator updates global model $\theta_{t+1} = \frac{1}{n} \sum_{i=1}^n \hat{q}_i^t$
  7:    Federator relays to client $j$ the other clients' indices $\{I_{i,\ell}^b\}_{\ell \in [n_{\mathrm{UL}}], b \in [B], i \in [n] \setminus \{j\}}$
  8:    **for** Clients $i \in [n]$ **do**
  9:       Client reconstructs from $\{I_{i,\ell}^b\}$ the global model $\theta_{t+1} = \frac{1}{n} \sum_{i=1}^n \hat{q}_i^t$
 10:    **end for**
 11:    Clients and federator set prior $p^t = \hat{\theta}_{t+1}$
 12:    $t \leftarrow t + 1$
 13: **until** Convergence

---

**Global Randomness** When global shared randomness is available, all clients can maintain the same priors at each iteration $t$, and, thereby, obtain the same global model estimates $\hat{\theta}_{i,t}$. The global model is known to the clients and the federator from initialization, and synchronization among all clients is ensured by choosing as prior $p_{i,u}^t = p_{i,d}^t$ the latest estimate of the global model $\hat{\theta}_{i,t}$. The clients utilize the globally shared randomness to sample the exact same samples from the same prior for uplink transmission at all iterations. Selected indices of such samples are transmitted to the federator to convey an estimate $\hat{q}_i^t$ of the posterior $q_i^t$, who reconstructs the global model $\theta_{t+1}$. Using the same prior in the downlink, i.e., the global model from the previous iteration, the updated model can be transmitted to the clients through importance sampling. Leveraging the shared randomness, all clients $i \in [n]$ sample from the same prior, and thus obtain the exact same estimate of the global model $\hat{\theta}_{i,t+1} = \hat{\theta}_{t+1}$, for all $i \in [n]$. Hence, we have that $p_{i,u}^t = p_{i,d}^t = \hat{\theta}_t$ for all $i \in [n]$.

In this version, the federator reconstructs the global model from estimates of the client posteriors $\hat{q}_i^t$. However, in the uplink, all clients sample from the same prior, which enables further improvements. Naively, the federator will reconstruct the global model using the indices $I_{i,\ell}^b$ for $b \in [B], \ell \in [n_{\mathrm{UL}}]$ received by the clients $i \in [n]$ through importance sampling, followed by an additional round of importance sampling for downlink transmission. Instead, and more efficiently, the federator can simply relay the indices to the respective other clients (i.e., client $j$ receives $I_{i,\ell}^b$ for $b \in [B], i \in [n] \setminus \{j\}, \ell \in [n_{\mathrm{UL}}]$), which reconstruct the same updated global model individually. This avoids introducing additional noise by a second round of compression and allows better convergence without additional communication costs facilitated by global randomness. We term this approach BICOMPFL-GR and summarize the procedure in Algorithm 2.

**Private Randomness** Without global randomness, maintaining the same prior among all clients is impossible without introducing additional communication. Instead, an additional round of importance sampling is needed for the downlink transmission, and each client obtains a different estimate of the global model $\hat{\theta}_{i,t}$ at each iteration. Hence, the clients' local trainings start from different estimates of the global model. In a non-stochastic setting, such a phenomenon has only been considered by Philippenko & Dieuleveut (2021); Gruntkowska et al. (2024). This raises the questions of the additional cost incurred due to lack of shared randomness in terms of both the convergence speed and the communication load and the choice of the priors $p_{i,u}^t$ and $p_{i,d}^t$.

For the uplink transmission of client $i$, any convex combination of $\hat{\theta}_{i,t}$ and $\hat{q}_i^t$ can be used as prior, i.e., $p_{i,u}^t = \lambda \hat{\theta}_{i,t} + (1 - \lambda)\hat{q}_i^t$, for some $0 \leq \lambda \leq 1$.[2] This is due to the availability of both quantities at the federator and client $i$. However, small $\lambda$ values are not expected to reduce the cost of communication reflected by $\mathrm{D_{KL}}\left(q_i^t \| p_{i,u}^t\right)$ since the prior global model estimate is likely to

---

[2]This adds a negligible cost of communication for the transmission of $\lambda$ if it is to be optimized at each round.

---

**Algorithm 3** BICOMPFL-PR with Private Shared Randomness

---

**Require:** Both clients and federator initialize the same global model $\mathbf{w}^{(0)}$ using a shared seed
**Ensure:** Clients set prior $\forall i \in [n] : p_{i,u}^t = p_{i,d}^t = \hat{\theta}_{i,0} = \hat{\theta}_0$
 1: **repeat**
 2:    **for** Clients $i \in [n]$ **do**
 3:       $q_i^t \leftarrow$ Local training of $\hat{\theta}_{i,t}$ according to Algorithm 1
 4:       Federator importance samples $n_{\text{UL}}$ masks $y_{i,\ell}^t \sim q_i^t$ with prior $p_{i,u}^t$
 5:       Federator estimates the client's posterior $\hat{q}_i^t = \frac{1}{n_{\text{UL}}} \sum_{\ell=1}^{n_{\text{UL}}} y_{i,\ell}^t$
 6:    **end for**
 7:    Federator updates global model $\theta_{t+1} = \frac{1}{n} \sum_{i=1}^n \hat{q}_i^t$
 8:    **for** Clients $i \in [n]$ **do**
 9:       Client importance samples $n_{\text{DL}}$ masks $x_{i,\ell}^t \sim \theta_{t+1}$ with prior $p_{i,d}^t$
10:       Client estimates global model: $\hat{\theta}_{i,t+1} = \frac{1}{n_{\text{DL}}} \sum_{\ell=1}^{n_{\text{DL}}} x_{i,\ell}^t$
11:       Clients set prior $p_{i,u}^t = p_{i,d}^t = \hat{\theta}_{i,t+1}$
12:    **end for**
13:    $t \leftarrow t + 1$
14: **until** Convergence

---

be similarly different from the posterior (in terms of the KL-divergence) than the former posterior estimate of the federator. Indeed, our numerical experiments have shown that the savings from choosing $\lambda \neq 1$, i.e., priors other than $\hat{\theta}_{i,t}$, are not significant. For simplicity, we thus propose to use $p_{i,u}^t = p_{i,d}^t = \hat{\theta}_{i,t}$. We term this approach BICOMPFL-PR and summarize the procedure in Algorithm 3. Choosing different priors is possible and only affects line 11 in Algorithm 3. We mention in passing that BICOMPFL-PR allows partial client participation, which is incompatible with shared randomness and the method BICOMPFL-GR.

**Block Allocation** We consider three different block allocation strategies: 1) fixed block size (referred to as "Fixed" in the experiments), where each block $b \in [B]$ is of the same size and constant across all $t$; 2) adaptive block allocation (Adaptive) as proposed by Isik et al. (2024), where each block size is separately optimized each iteration $t$; and 3) adaptive average allocation (Adaptive-Avg), where the block sizes are equal but optimized at each iteration $t$ according to the average KL-divergence per block. We refer the reader to Appendix C for a detailed discussion on this.

## 4 NUMERICAL EXPERIMENTS

We conducted experiments to evaluate the performance of our proposed BICOMPFL-GR and BICOMPFL-PR schemes, and compare against baseline FL strategies without compression (FedAvg or PSGD) McMahan et al. (2017) and several non-stochastic bi-directional compression schemes that employ different combinations of compression, error-feedback, and momentum. In particular, we compare against DOUB-LESQUEEZE (Tang et al., 2019), MEM-SGD (Stich et al., 2018), NEOLITHIC (Huang et al., 2022), CSER (Xie et al., 2020), and the recently proposed LIEC (Cheng et al., 2024). SignSGD (Seide et al., 2014) serves to compress the transmitted gradients for all the schemes. We further compare with M3 (Gruntkowska et al., 2024), which partitions the model into disjoint parts for downlink transmission and transmits to each client a different part of the model. While the paper was focused on RandK compression for the uplink (i.e., transmitting random $K$ entries of the gradient), we use TopK

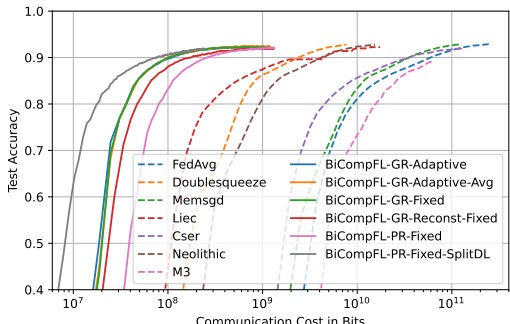

Figure 1: Fashion MNIST 4CNN i.i.d.

(Wangni et al., 2018; Shi et al., 2019), which we found to achieve much more stable results.

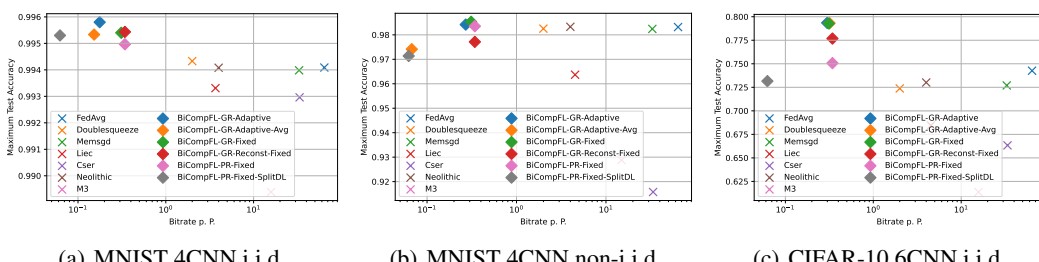

(a) MNIST 4CNN i.i.d.    (b) MNIST 4CNN non-i.i.d.    (c) CIFAR-10 6CNN i.i.d.

Figure 2: Maximum test accuracy as a function of the total communication cost measured as the bitrate per parameter.

We study the setting of $n = 10$ clients collaboratively training a convolutional neural network (CNN)-based classifier for the datasets MNIST, Fashion-MNIST and CIFAR-10 under the orchestration of a federator. For MNIST, we use two different models, LeNet-5 (Lecun et al., 1998) and a 4-layer convolutional neural network (4CNN) proposed by Ramanujan et al. (2020). The latter is also used to train on Fashion MNIST. For CIFAR-10, we use a larger neural network with 6 convolutional layers (6CNN). We train MNIST and Fashion-MNIST for 200 epochs and CIFAR-10 for 400 global iterations. Consistently through all experiments and datasets, we carry $L = 3$ local iterations per client per global iteration. We evaluate the performance of the schemes in two different settings: with uniform data allocation (i.i.d.) to model homogeneous systems and a non-i.i.d. setting to model heterogeneous systems, where data allocation for each client is drawn from a Dirichlet distribution with parameter $\alpha = 0.1$. This is considered a rather challenging regime due to high-class imbalance. Every result shows the average across three simulation runs with different seeds. Details on the simulation setup and the network architectures are deferred to Appendix D. Consistently throughout all experiments, our proposed methods provide order-wise improvements in the communication cost while achieving state-of-the art accuracies.

We plot in Fig. 1 the test accuracies for all the schemes as a function of the total communication cost in bits per parameter and per epoch. While all the schemes achieve approximately the same maximum test accuracy, BiCOMPFL-GR and BiCOMPFL-PR require substantially less communication. Hence, when the bandwidths of uplink and downlink transmissions are limited, both variations of the proposed method achieve better test accuracies. Turning our focus to the different variations of our scheme, it can be observed that, without partitioning the model for downlink compression, BiCOMPFL-PR convergences significantly slower than BiCOMPFL-GR for any block allocation method. This highlights the intuition above that the additional importance sampling step in downlink incurs further noise, which reduces the convergence speed. However, when we partition the model in the downlink and only send disjoint parts to each client through importance sampling (BiCOMPFL-PR-Fixed-SplitDL), the downlink communication cost reduces by a factor of $n$. In the regime of Fashion MNIST with uniform data allocation, this comes without performance degradation, and is hence the method of choice in this regime. We additionally simulated BiCOMPFL-GR with the suboptimal implementation (BiCOMPFL-GR-Reconst-Fixed), in which the federator first reconstructs the global model, and then performs an additional importance sampling step for downlink transmission. This naturally reduces the convergence speed per iteration without gains in the communication cost. Hence, justifying the choice of BiCOMPFL-GR.

We plot in Fig. 2(a) the average bitrate of each scheme over the maximum test accuracy for MNIST and 4CNN. The average bitrate is by more than a factor of 1000 less than the baseline FedAvg, and more than a **factor of 32 less** than DOUBLESQUEEZE, NEOLITHIC and LIEC, which perform best among the non-stochastic bi-directional compression methods.

We perform the same study for non-i.i.d. data allocation according to a Dirichlet distribution with parameter $\alpha = 0.1$, and show the maximum test accuracies over the average bitrate in Fig. 2(b). It can be found that partitioning the model in BiCOMPFL-PR worsens the final accuracy of the model. While the model converges faster, it does not achieve the same accuracies as BiCOMPFL-GR and BiCOMPFL-PR without partitioning. This hints towards hybrid schemes for BiCOMPFL-PR, where the training begins with partitioning on the downlink, and the scheme later switches to full transmission.

In Fig. 2(c), we provide the results for CIFAR-10 and uniform data allocation. BiCompFL-GR and BiCompFL-PR both achieve better results with a bitrate **smaller by a factor of 5** than the best baselines. More detailed simulation results can be found in the Appendices.

The adaptive block allocation (Adaptive) of Isik et al. (2024) saves communication costs in many settings and provides better performance than the fixed block allocation (Fixed), due to more accurate importance sampling tailored to the exact divergences. The proposed low complexity adaptive strategy based on the average KL-divergence (Adaptive-Avg) per block can additionally save in communication (and computation) with no or little performance degradation. We refer the reader to Appendix E for further extensive experiments.

## 5 THEORETICAL RESULTS

The exact dynamics of the system over time are challenging to analyze due to the round-dependent interplay of stochastic FL with the transmission noise; we, hence, focus on a specific iteration $t$ and comment on the inter-round dependency later. When the latest global model estimate $\hat{\theta}_{i,t}$ is chosen as a prior in importance sampling, the cost of communication on the uplink is mainly determined by how far the model evolves during the client's training, i.e., $\mathrm{D_{KL}}(q_i^t \| p_{i,u}^t) = \mathrm{D_{KL}}(q_i^t \| \hat{\theta}_{i,t})$. After communicating samples of the posteriors, the federator obtains an estimate $\hat{q}_i^t$ for all $i \in [n]$. The cost of communication on the downlink to client $i$ is then determined by $\mathrm{D_{KL}}(\frac{1}{n} \sum_{i=1}^n \hat{q}_i^t \| \hat{\theta}_{i,t})$. While $\mathrm{D_{KL}}(q_i^t \| \hat{\theta}_{i,t})$ depends on the progress during client training, the core challenge is to bound the expected KL-divergence of each model estimate $\mathrm{D_{KL}}(\hat{q}_i^t \| \hat{\theta}_{i,t})$ in the presence of potentially different priors, i.e., $\hat{\theta}_{i,t} \neq \hat{\theta}_{j,t}, i \neq j$. For each client $i$, the overall communication cost is in the order of

$$n_{\mathrm{DL}} \exp\left(\mathrm{D_{KL}}\left(\frac{1}{n} \sum_{i=1}^n \hat{q}_i^t \| p_{i,d}^t\right)\right) + n_{\mathrm{UL}} \exp\left(\mathrm{D_{KL}}\left(q_i^t \| p_{i,u}^t\right)\right).$$

We will next quantify $\mathrm{D_{KL}}(\frac{1}{n} \sum_{i=1}^n \hat{q}_i^t \| \hat{\theta}_{i,t})$ for the case $p_{i,u}^t = p_{i,d}^t$, however, the analysis can be extended to $p_{i,u}^t \neq p_{i,d}^t$ by an additional assumption on the divergence between the two priors.

For the theoretical analysis, we focus on the scalar case for a single iteration $t$, where the client $i \in [n]$ has a posterior $Q_i$ (also: the client's local model), and the federator and the client $i$ share a common prior $P_i$, both are Bernoulli distributions with parameters $q_i$ and $p_i$, respectively. In the context of FL, the client locally trains $P_i$ and results with $Q_i$. According to Chatterjee & Diaconis (2018) and the multi-client extension of Isik et al. (2024), the communication cost in the uplink is determined by $\exp(\mathrm{D_{KL}}(Q_i \| P_i))$. After uplink transmission, the federator obtains an estimate $\hat{q}_i$ of $q_i$; and hence, the updated global model is given by $\frac{1}{n} \sum_{i=1}^n \hat{q}_i$. The communication cost in the downlink for client $i$ is determined by $\mathrm{d_{KL}}\left(\frac{1}{n} \sum_{i=1}^n \hat{q}_i \| p_i\right)$. Our theoretical contribution is a new high probability upper bound on this quantity, which refines previous importance sampling analysis, for the special case of Bernoulli distributions. Let $X$ be a Bernoulli sample obtained through importance sampling. As an initial step, we derive an upper bound on the difference between $q_i$ and the probability $\Pr(X) = 1$ that the samples are drawn from, which vanishes when $p_i = q_i$ (and hence $\mathrm{d_{KL}}(q_i \| p_i) = 0$). We note that the bound of (Chatterjee & Diaconis, 2018, Theorem 1.1) does not saitsfy this natural property. We formally state the result in Proposition 1 in Appendix A, which, however, does not yet capture the dependency on the number of samples $n_{\mathrm{IS}}$ used in importance sampling to sample an index. We refine Proposition 1 with Lemma 1 (cf. Appendix A), which additionally captures this dependency, and will allow us to derive an upper bound on $\mathrm{d_{KL}}\left(\frac{1}{n} \sum_{i=1}^n \hat{q}_i \| p_i\right)$. Lemma 1 is of independent interest and can be seen as a refinement of the analysis by Chatterjee & Diaconis (2018) for Bernoulli distributions. It is required to prove Theorem 1.

For the statement of the following theorem, we assume that the progress by one local client training is bounded by $|q_j - p_j| \leq \rho$ for all $j \in [n]$. Using Pinsker's inequality to bound $|q_j - p_j| \leq \frac{1}{2}\sqrt{\mathrm{d_{KL}}(q_j \| p_j)/2}$, this is a natural assumption given from the KL-proximity term of mirror descent (for one local iteration), and can be strictly enforced through the projection of $q_j$ onto a KL ball around $p_j$ of fixed divergence. We further assume that the difference between the clients' priors, i.e., their global model estimates in our algorithms, are bounded as $|p_i - p_j| \leq \zeta$ for all $i, j \in [n]$.

**Theorem 1.** *Assume $p_j > \zeta$ for all $j \in [n]$, for $\Delta_j := \frac{q_j}{p_j - \zeta} - \frac{1 - q_j}{1 - p_j + \zeta}$ and $\Delta_j' := q_j \left( \frac{p_j + \zeta}{q_j} + \frac{1 - p_j + \zeta}{1 - q_j} \right)$, with probability $1 - \delta'$, the global model divergence $d_{KL}(\frac{1}{n} \sum_{j=1}^n \hat{q}_j || p_i)$ is upper bounded by*

$$\sum_{j=1}^n \frac{2}{n \min\{p_i, 1 - p_i\}} \left( \frac{\Delta_j'}{n_{IS}^2} + \mathcal{O} \left( (\Delta_j + \Delta_j^2) \sqrt{\frac{6(p_i + \zeta) \log{(2n_{IS})}}{n_{IS}}} \right) + \sqrt{\frac{\ln(2/\delta')}{2n_{UL}}} + \rho + \zeta \right)^2.$$

By Chatterjee & Diaconis (2018), this provides an immediate bound on the cost of downlink transmission. The bound applies to both algorithms BICOMPFL-PR and BICOMPFL-GR. However, when all priors $p_j$ are the same (such as in BICOMPFL-GR-Reconst), i.e., $\zeta = 0$, the bound simplifies accordingly. The explicit dependency on the factor $1/\sqrt{n_{UL}}$ reflects the interplay between uplink and downlink cost. The parameter $\zeta$ gives rise to an inter-round dependency of the communication cost. The more accurate the estimation of the global model in the previous iteration (given the priors are chosen as $\hat{\theta}_{i,t}$), the smaller $\zeta$, and hence the lower the transmission cost in the subsequent iteration. The proofs of Proposition 1, Lemma 1, and Theorem 1 can be found in Appendix A.

## 6 RELATED WORK

Followed by the introduction of FL by McMahan et al. (2017), lossy compression of gradients or model updates has been a long studied narrative in FL, with prominent representatives such as SignSGD, also known as 1-bit Stochastic Gradient Descent (SGD) (Seide et al., 2014), QSGD (Alistarh et al., 2017), TernGrad (Wen et al., 2017), SignSGD with error feedback (Karimireddy et al., 2019), vector-quantized SGD (Gandikota et al., 2021) and natural compression (Horvóth et al., 2022). Such methods retain satisfactory final model accuracy even with aggressive quantization. Sparsification-based methods have also been considered as alternatives, e.g., TopK (Wangni et al., 2018; Shi et al., 2019). The importance of bi-directional gradient compression in many settings was outlined by Philippenko & Dieuleveut (2020). Many schemes were proposed that leverage combinations of gradient compression in the uplink and downlink, error-feedback, and momentum, e.g., Mem-SGD (Stich et al., 2018), DoubleSqueeze (Tang et al., 2019), block-wise SignSGD with momentum (Zheng et al., 2019), communication-efficient SGD with error reset (CSER) (Xie et al., 2020), Artemis (Philippenko & Dieuleveut, 2020), Neolithic (Huang et al., 2022), DOCOFL (Dorfman et al., 2023), EF21-P and friends (Gruntkowska et al., 2023), 2Direction (Tyurin & Richtárik, 2023), M3 Gruntkowska et al. (2024), and LIEC (Cheng et al., 2024). With the exception of the methods MCM (Philippenko & Dieuleveut, 2021) and M3 (Gruntkowska et al., 2024), each client receives the same broadcast, potentially compressed, global gradient or model update. Isik et al. (2024) studied uplink compression for stochastic FL and showed significant communication reduction with competitive performance. Their framework, termed KLMS, applies to a variety of stochastic compressors and to Bayesian FL settings, e.g., QLSD Vono et al. (2022). The compression is based on importance sampling, thoroughly studied by Chatterjee & Diaconis (2018). Such methods, known as relative entropy coding (REC), have been used in FL in conjunction with differential privacy, cf. DP-REC (Triastcyn et al., 2022).

Since the lottery ticket hypothesis (Frankle & Carbin, 2019) a variety of works were concerned with finding sparse subnetworks of neural networks that achieve satisfactory accuracy. Ramanujan et al. (2020) showed that randomly weighted networks contain suitable subnetworks of large neural networks capable of achieving competitive performance. Isik et al. (2023) formulated a probabilistic method of training neural network masks collaboratively in an FL context.

## 7 CONCLUSION

In this paper, we illuminated the problem of bi-directional compression in stochastic FL using the specific instance of federated probabilistic mask training. By leveraging side-information through carefully chosen prior distributions, the total communication costs can be reduced by factors between 5 and 32 compared to non-stochastic FL baselines while achieving state-of-the-art accuracies on classification tasks in both homogeneous and heterogeneous FL regimes. We thereby close the gap of downlink compression for stochastic FL and complement the existing literature on bi-directional compression for standard FL. By allowing different priors among all clients, this work opens the door to studying compression under side-information in *decentralized stochastic FL*, where a central coordinator is missing. Our theoretical results are of independent interest and may find application in various scenarios where importance sampling is used.

## 8 REPRODUCIBILITY STATEMENT

In addition to the algorithmic details and the clients' training procedure function (cf. Algorithms 1 to 3), we provide in Section 4 the most important hyperparameters used in our experiments, such as local and global iterations, and data allocation. Further parameter information, such as batch size, learning rates and the choice of the optimizer can be found in Appendix E, together with details on the neural network architectures and the hardware cluster used for running the experiments. Particularities of the block allocation required for the operation of our schemes are described in Appendix C. All assumptions required for the theoretical analysis are stated in Section 5. Full proofs of all claims, including formal statements, can be found in Appendix A.

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

## A  PROOFS AND INTERMEDIATE RESULTS

In the following, we provide the formal statements of Proposition 1 and Lemma 1 including their proofs. Parts of the proof of Proposition 1 will be used to prove Lemma 1. We prove Theorem 1 afterward.

**Proposition 1.** *For a sample $X_\ell$ transmitted by importance sampling with posterior and prior Bernoulli distributions with parameters $q$ and $p$, we have*

$$| \Pr(X_\ell = 1) - q| \leq q \left( \max \left\{ \frac{p}{q}, \frac{1-p}{1-q}, \frac{q}{p}, \frac{1-q}{1-p} \right\} - 1 \right).$$

*Proof of Proposition 1.* Assume a party wants to sample from a Bernoulli distribution $Q$ with parameter $q$, which is held by another party. Both parties share a common prior $P$ in the form of a Bernoulli distribution with parameter $p$ and have access to shared randomness. Fix any sample index $\ell$ for the moment (this index will be needed for the proof of Theorem 1). Both parties sample $Kn_{\text{IS}}$ i.i.d. samples $X_{\ell,i} \sim P$ for $i \in [n_{\text{IS}}]$ independently and identically from $P$. The party holding $Q$ constructs an auxiliary distribution

$$W_\ell(i) = \frac{Q(X_{\ell,i})/P(X_{\ell,i})}{\sum_{i=1}^{n_{\text{IS}}} Q(X_{\ell,i})/P(X_{\ell,i})},$$

from which it samples to obtain an index $I_\ell$. The index is transmitted to the other party, which reconstructs the corresponding sample $X_{\ell,I_\ell}$.

To bound the difference $| \Pr(X_\ell = 1) - q|$, i.e., the target Bernoulli parameter compared to the parameter which the sample is drawn from, by the independence of the samples $X_{\ell,I_\ell}$ for different $\ell$, we focus on a single sample $\ell \in [K]$, for which it holds that

$$\Pr(X_{\ell,I_\ell} = 1)$$

$$= \sum_{i=1}^{n_{\text{IS}}} \sum_{\{x_1,\dots,x_{n_{\text{IS}}}:x_i=i\}} \Pr(X_{\ell,1} = x_1, \dots, X_{\ell,n_{\text{IS}}} = x_{n_{\text{IS}}}) \Pr(I_\ell = i \mid X_{\ell,1} = x_1, \dots, X_{\ell,n_{\text{IS}}} = x_{n_{\text{IS}}})$$

$$\overset{(a)}{=} n_{\text{IS}} \sum_{\{x_2,\dots,x_{n_{\text{IS}}}\}} \Pr(X_{\ell,1} = 1, X_{\ell,2} = x_2, \dots, X_{\ell,n_{\text{IS}}} = x_{n_{\text{IS}}})$$

$$\cdot \Pr(I_\ell = 1 \mid X_{\ell,1} = 1, X_{\ell,2} = x_2, \dots, X_{\ell,n_{\text{IS}}} = x_{n_{\text{IS}}})$$

$$\overset{(b)}{=} n_{\text{IS}} \sum_{\text{L}=0}^{n_{\text{IS}}-1} \sum_{\{x_2,\dots,x_{n_{\text{IS}}}:\sum_{i=2}^{n_{\text{IS}}}=\text{L}\}} \Pr(X_{\ell,1} = 1, X_{\ell,2} = x_2, \dots, X_{\ell,n_{\text{IS}}} = x_{n_{\text{IS}}})$$

$$\cdot \Pr(I_\ell = 1 \mid X_{\ell,1} = 1, X_{\ell,2} = x_2, \dots, X_{\ell,n_{\text{IS}}} = x_{n_{\text{IS}}}),$$

where $(a)$ follows from symmetry, $(b)$ follows since by permutation invariance, the inner probability only depends on the number of ones in $\{x_2, \dots, x_{n_{\text{IS}}}\}$.

The inner probability is given by the distribution $W_\ell(i)$. Given that $X_{\ell,1} = 1$ and that $\sum_{i=2}^{n_{\text{IS}}} X_{\ell,\ell} = \text{L}$, it holds that

$$\sum_{i=1}^{n_{\text{IS}}} Q(X_{\ell,i})/P(X_{\ell,i}) = (\text{L}+1) \cdot \frac{q}{p} + (n_{\text{IS}} - \text{L} - 1) \cdot \frac{1-q}{1-p}.$$

Hence,

$$\Pr(I_\ell = 1 \mid X_{\ell,1} = 1, X_{\ell,2} = x_2, \dots, X_{\ell,n_{\text{IS}}} = x_{n_{\text{IS}}}) = \frac{\frac{q}{p}}{(\text{L}+1) \cdot \frac{q}{p} + (n_{\text{IS}} - \text{L} - 1) \cdot \frac{1-q}{1-p}},$$

which is independent of the exact choice of $\{x_2, \dots, x_{n_{\text{IS}}}\}$ given their sum $\sum_{i=2}^{n_{\text{IS}}} X_{\ell,i} = \text{L}$. Since $\Pr(X_{\ell,1} = 1, X_{\ell,2} = x_2, \dots, X_{\ell,n_{\text{IS}}} = x_{n_{\text{IS}}}) = p^{\text{L}+1}(1-p)^{n_{\text{IS}}-\text{L}-1}$ by the Bernoulli distribution assumption, we have

$$\Pr(X_{\ell,I_\ell} = 1) = n_{\text{IS}} \sum_{\text{L}=0}^{n_{\text{IS}}-1} \binom{n_{\text{IS}} - 1}{\text{L}} p^{\text{L}+1}(1-p)^{n_{\text{IS}}-\text{L}-1} \frac{\frac{q}{p}}{(\text{L}+1) \cdot \frac{q}{p} + (n_{\text{IS}} - \text{L} - 1) \cdot \frac{1-q}{1-p}},$$

Defining a binary random variable M with sample space $\left\{\frac{q}{p}, \frac{1-q}{1-p}\right\}$, for a Bernoulli distribution $\text{Ber}\left(\frac{L+1}{n_{\text{IS}}}\right)$ with success probability parameter $\frac{L+1}{n_{\text{IS}}}$, where a success refers to the outcome $M = \frac{q}{p}$, we can write that

$$
\begin{aligned}
\Pr(X_{\ell, I_\ell} = 1) &= q \cdot \sum_{L=0}^{n_{\text{IS}}-1} \binom{n-1}{L} p^{L}(1-p)^{n_{\text{IS}}-L-1} \frac{1}{\frac{L+1}{n_{\text{IS}}} \frac{q}{p} + \frac{n_{\text{IS}}-L-1}{n_{\text{IS}}} \frac{1-q}{1-p}} \\
&= q \cdot \mathbb{E}\left[\frac{1}{\frac{L+1}{n_{\text{IS}}} \frac{q}{p} + \frac{n_{\text{IS}}-L-1}{n_{\text{IS}}} \frac{1-q}{1-p}}\right] = q\mathbb{E}\left[\frac{1}{\mathbb{E}_{\text{Ber}\left(\frac{L+1}{n_{\text{IS}}}\right)}[M]}\right] \\
&\overset{(a)}{\leq} q\mathbb{E}\left[\mathbb{E}_{\text{Ber}\left(\frac{L+1}{n_{\text{IS}}}\right)}\left[\frac{1}{M}\right]\right],
\end{aligned}
\tag{1}
$$

where the outer expectation is over the binomial distribution with $n_{\text{IS}} - 1$ trials and success probability $p$, i.e., $L \sim \text{Binomial}(n_{\text{IS}} - 1, p)$, and where $(a)$ follows from Jensen's inequality over the inner expectation. Hence,

$$
\begin{aligned}
\Pr(X_{\ell, I_\ell} = 1) - q &= q\left(\frac{\Pr(X_{\ell, I_\ell} = 1)}{q} - 1\right) \\
&\leq q\left(\mathbb{E}\left[\mathbb{E}_{\text{Ber}\left(\frac{L+1}{n_{\text{IS}}}\right)}\left[\frac{1}{M}\right]\right] - 1\right)
\end{aligned}
\tag{2}
$$

Since $\frac{1}{\mathbb{E}_{\text{Ber}\left(\frac{L+1}{n_{\text{IS}}}\right)}[M]} \geq 2 - \mathbb{E}_{\text{Ber}\left(\frac{L+1}{n_{\text{IS}}}\right)}[M]$, it also follows from (1) that

$$
\begin{aligned}
\Pr(X_{\ell, I_\ell} = 1) &= q \cdot \mathbb{E}\left[\frac{1}{\frac{L+1}{n_{\text{IS}}} \frac{q}{p} + \frac{n_{\text{IS}}-L-1}{n_{\text{IS}}} \frac{1-q}{1-p}}\right] = q\mathbb{E}\left[\frac{1}{\mathbb{E}_{\text{Ber}\left(\frac{L+1}{n_{\text{IS}}}\right)}[M]}\right] \\
&\geq q\mathbb{E}\left[2 - \mathbb{E}_{\text{Ber}\left(\frac{L+1}{n_{\text{IS}}}\right)}[M]\right],
\end{aligned}
$$

from which we have

$$
\Pr(X_{\ell, I_\ell} = 1) - q \geq q\left(1 - \mathbb{E}\left[\mathbb{E}_{\text{Ber}\left(\frac{L+1}{n_{\text{IS}}}\right)}[M]\right]\right).
\tag{3}
$$

Combining the upper and lower bound in (2) and (3), respectively, we derive

$$
\begin{aligned}
|\Pr(X_{\ell, I_\ell} = 1) - q| &\leq q\left(\max\left\{\mathbb{E}\left[1 - \mathbb{E}_{\text{Ber}\left(\frac{L+1}{n_{\text{IS}}}\right)}[M]\right], \mathbb{E}\left[\mathbb{E}_{\text{Ber}\left(\frac{L+1}{n_{\text{IS}}}\right)}\left[\frac{1}{M}\right]\right]\right\} - 1\right) \\
&\leq q\left(\mathbb{E}\left[\max\left\{\mathbb{E}_{\text{Ber}\left(\frac{L+1}{n_{\text{IS}}}\right)}[M], \mathbb{E}_{\text{Ber}\left(\frac{L+1}{n_{\text{IS}}}\right)}\left[\frac{1}{M}\right]\right\}\right] - 1\right) \\
&\leq q\left(\mathbb{E}\left[\mathbb{E}_{\text{Ber}\left(\frac{L+1}{n_{\text{IS}}}\right)}\left[\max\left\{M, \frac{1}{M}\right\}\right]\right] - 1\right) \\
&\leq q\left(\mathbb{E}\left[\max\left\{\frac{p}{q}, \frac{1-p}{1-q}, \frac{q}{p}, \frac{1-q}{1-p}\right\}\right] - 1\right) \\
&= q\left(\max\left\{\frac{p}{q}, \frac{1-p}{1-q}, \frac{q}{p}, \frac{1-q}{1-p}\right\} - 1\right).
\end{aligned}
$$

This concludes the proof. $\qquad\square$

**Lemma 1.** *For a sample $X_\ell$ transmitted via importance sampling with posterior and prior being Bernoulli distributions with parameters $q$ and $p$, $\Delta := \frac{q}{p} - \frac{1-q}{1-p}$ and $\Delta' := q\left(\frac{p}{q} + \frac{1-p}{1-q}\right)$, we have*

$$
|\Pr(X_\ell = 1) - q| \leq \frac{\Delta'}{n_{IS}^2} + \mathcal{O}\left((\Delta + \Delta^2)\sqrt{\frac{6p\log(2n_{IS})}{n_{IS}}}\right).
$$

*Proof of Lemma 1.* The proof starts with the same derivations as for the proof of Proposition 1, which we follow until (1) to get

$$\Pr(X_{\ell,I_\ell} = 1) = q\mathbb{E}\left[\frac{1}{\mathbb{E}_{\text{Ber}\left(\frac{L+1}{n_{\text{IS}}}\right)}[\text{M}]}\right]$$

Since L is a random quantity that follows a Binomial distribution, we bound $|\Pr(X_{\ell,I_\ell} = 1) - q|$ using a concentration bound on L. The relative (multiplicative) Chernoff bound states that

$$\Pr(|\text{L} - \varepsilon(n_{\text{IS}}p)| \geq \varepsilon n_{\text{IS}}p) = \Pr(\text{L} - \varepsilon(n_{\text{IS}}p) \geq \varepsilon n_{\text{IS}}p) + \Pr(\text{L} - \varepsilon(n_{\text{IS}}p) \leq -\varepsilon n_{\text{IS}}p)$$

$$\leq 2\exp\left(-\frac{\varepsilon^2 n_{\text{IS}}p}{3}\right)$$

for any $\varepsilon \in [0, 1]$. Setting $\varepsilon = \sqrt{\frac{3\log(2/\delta)}{n_{\text{IS}}p}}$ implies that

$$|\text{L} - n_{\text{IS}}p| \geq \sqrt{3n_{\text{IS}}p\log(2/\delta)}$$

with probability at most $\delta$. Setting $\delta = \frac{1}{n_{\text{IS}}^2}$, we obtain for a concentration parameter[3] $\eta_\delta := \sqrt{\frac{6p\log(2n_{\text{IS}})}{n_{\text{IS}}}}$ that

$$\mathcal{E} := \{|\text{L} - n_{\text{IS}}p| \geq n_{\text{IS}}\eta_\delta\}$$

with probability $\Pr(\mathcal{E}) \leq \frac{1}{n_{\text{IS}}^2}$.

Then, we can write

$$\Pr(X_{\ell,I_\ell} = 1) = q\mathbb{E}\left[\frac{1}{\mathbb{E}_{\text{Ber}\left(\frac{L+1}{n_{\text{IS}}}\right)}[\text{M}]}\right]$$

$$= q\mathbb{E}\left[\frac{1}{\mathbb{E}_{\text{Ber}\left(\frac{L+1}{n_{\text{IS}}}\right)}[\text{M}]} \cdot \mathbb{1}\{\mathcal{E}^c\}\right] + q\mathbb{E}\left[\frac{1}{\mathbb{E}_{\text{Ber}\left(\frac{L+1}{n_{\text{IS}}}\right)}[\text{M}]} \cdot \mathbb{1}\{\mathcal{E}\}\right] \tag{4}$$

Assume for now that $q < p$ (we will later proof the opposite event), then $\frac{1}{\mathbb{E}_{\text{Ber}\left(\frac{L+1}{n_{\text{IS}}}\right)}[\text{M}]}$ is strictly non-increasing in L since $\frac{q}{p} < \frac{1-q}{1-p}$, and hence, when $\mathcal{E}^c$ holds and hence L concentration around the average that

$$\frac{1}{\mathbb{E}_{\text{Ber}\left(\frac{L+1}{n_{\text{IS}}}\right)}[\text{M}]} \leq \frac{1}{\mathbb{E}_{\text{Ber}\left(\frac{(L+1)\cdot(p-\eta_\delta)}{n_{\text{IS}}}\right)}[\text{M}]}$$

$$= \frac{1}{\frac{(n_{\text{IS}}-1)(p-\eta_\delta)+1}{n_{\text{IS}}}\frac{q}{p} + \frac{n_{\text{IS}}-1-(n_{\text{IS}}-1)(p-\eta_\delta)}{n_{\text{IS}}}\frac{1-q}{1-p}}$$

$$= \frac{1}{\left(p - \frac{p}{n_{\text{IS}}} + \frac{\eta_\delta}{n_{\text{IS}}} - \eta_\delta + \frac{1}{n_{\text{IS}}}\right)\frac{q}{p} + \left(1 - p - \frac{1}{n_{\text{IS}}} + \frac{p}{n_{\text{IS}}} + \eta_\delta - \frac{\eta_\delta}{n_{\text{IS}}}\right)\frac{1-q}{1-p}}$$

$$= \frac{1}{1 + \left(\frac{q}{p} - \frac{1-q}{1-p}\right)\left(\frac{1-p+\eta_\delta-n\eta_\delta}{n_{\text{IS}}}\right)}$$

$$= 1 + \sum_{\kappa=1}^{\infty}(-1)^\kappa\left(\frac{q}{p} - \frac{1-q}{1-p}\right)^\kappa\left(\frac{1-p+\eta_\delta-n\eta_\delta}{n_{\text{IS}}}\right)^\kappa,$$

---

[3]Note that we can assume $p + \eta_\delta \leq 1$ and $p - \eta_\delta \geq 0$, otherwise the concentration can be trivially bounded.

where the last step is by Taylor expansion. Using (4) and the monotonicity of $\frac{1}{\mathbb{E}_{\mathrm{Ber}\left(\frac{L+1}{n_{\mathrm{IS}}}\right)}[\mathrm{M}]}$, we write

$$
\mathrm{Pr}(X_{\ell,I_\ell} = 1) = q\mathbb{E}\left[\frac{1}{\mathbb{E}_{\mathrm{Ber}\left(\frac{L+1}{n_{\mathrm{IS}}}\right)}[\mathrm{M}]}\right]
$$

$$
\leq q\left(1 + \sum_{\kappa=1}^{\infty}(-1)^\kappa\left(\frac{q}{p} - \frac{1-q}{1-p}\right)^\kappa\left(\frac{1-p+\eta_\delta-n\eta_\delta}{n_{\mathrm{IS}}}\right)^\kappa\right) + q\delta\frac{p}{q},
$$

and hence

$$
\mathrm{Pr}(X_{\ell,I_\ell} = 1) - q \leq \delta p + (1-\delta)\sum_{\kappa=1}^{\infty}(-1)^\kappa\left(\frac{q}{p} - \frac{1-q}{1-p}\right)^\kappa\left(\frac{1-p+\eta_\delta-n\eta_\delta}{n_{\mathrm{IS}}}\right)^\kappa
$$

Similarly, we get by bounding $\frac{1}{\mathbb{E}_{\mathrm{Ber}\left(\frac{L+1}{n_{\mathrm{IS}}}\right)}[\mathrm{M}]} \geq \frac{1}{\mathbb{E}_{\mathrm{Ber}\left(\frac{(L+1)\cdot(p+\eta_\delta)}{n_{\mathrm{IS}}}\right)}[\mathrm{M}]}$ and using (4) that

$$
\mathrm{Pr}(X_{\ell,I_\ell} = 1) - q \geq \delta q\frac{1-p}{1-q} + (1-\delta)\sum_{\kappa=1}^{\infty}(-1)^\kappa\left(\frac{q}{p} - \frac{1-q}{1-p}\right)^\kappa\left(\frac{1-p-\eta_\delta+n\eta_\delta}{n_{\mathrm{IS}}}\right)^\kappa \Leftrightarrow
$$

$$
q - \mathrm{Pr}(X_{\ell,I_\ell} = 1) \leq -\delta q\frac{1-p}{1-q} + (1-\delta)\sum_{\kappa=1}^{\infty}(-1)^{\kappa+1}\left(\frac{q}{p} - \frac{1-q}{1-p}\right)^\kappa\left(\frac{1-p-\eta_\delta+n\eta_\delta}{n_{\mathrm{IS}}}\right)^\kappa.
$$

When $p \leq q$, then $\frac{1}{\mathbb{E}_{\mathrm{Ber}\left(\frac{L+1}{n_{\mathrm{IS}}}\right)}[\mathrm{M}]}$ is strictly non-decreasing, hence, under $\mathcal{E}$, we have

$$
\frac{1}{\mathbb{E}_{\mathrm{Ber}\left(\frac{L+1}{n_{\mathrm{IS}}}\right)}[\mathrm{M}]} \leq \frac{1}{\mathbb{E}_{\mathrm{Ber}\left(\frac{(L+1)\cdot(p+\eta_\delta)}{n_{\mathrm{IS}}}\right)}[\mathrm{M}]} = 1 + \sum_{\kappa=1}^{\infty}(-1)^\kappa\left(\frac{q}{p} - \frac{1-q}{1-p}\right)^\kappa\left(\frac{1-p-\eta_\delta+n\eta_\delta}{n_{\mathrm{IS}}}\right)^\kappa,
$$

and thus from (4) that

$$
\mathrm{Pr}(X_{\ell,I_\ell} = 1) - q \leq q\delta\frac{1-p}{1-q} + (1-\delta)\sum_{\kappa=1}^{\infty}(-1)^\kappa\left(\frac{q}{p} - \frac{1-q}{1-p}\right)^\kappa\left(\frac{1-p-\eta_\delta+n\eta_\delta}{n_{\mathrm{IS}}}\right)^\kappa.
$$

Similarly, we bound $\frac{1}{\mathbb{E}_{\mathrm{Ber}\left(\frac{L+1}{n_{\mathrm{IS}}}\right)}[\mathrm{M}]} \leq \frac{1}{\mathbb{E}_{\mathrm{Ber}\left(\frac{(L+1)\cdot(p+\eta_\delta)}{n_{\mathrm{IS}}}\right)}[\mathrm{M}]}$ to obtain

$$
\mathrm{Pr}(X_{\ell,I_\ell} = 1) - q \geq q\delta\frac{p}{q} + (1-\delta)\sum_{\kappa=1}^{\infty}(-1)^\kappa\left(\frac{q}{p} - \frac{1-q}{1-p}\right)^\kappa\left(\frac{1-p+\eta_\delta-n\eta_\delta}{n_{\mathrm{IS}}}\right)^\kappa \Leftrightarrow
$$

$$
q - \mathrm{Pr}(X_{\ell,I_\ell} = 1) \leq -q\delta\frac{p}{q} + (1-\delta)\sum_{\kappa=1}^{\infty}(-1)^{\kappa+1}\left(\frac{q}{p} - \frac{1-q}{1-p}\right)^\kappa\left(\frac{1-p+\eta_\delta-n\eta_\delta}{n_{\mathrm{IS}}}\right)^\kappa
$$

Since $0 \leq p + \eta_\delta \leq 1$ and $1 \geq p - \eta_\delta \geq 0$ by an appropriate choice of the concentration intervals, we have by approximations up to second order terms that

$$
|\mathrm{Pr}(X_{\ell,I_\ell} = 1) - q| \leq q\delta\max\left\{\frac{p}{q}, \frac{1-p}{1-q}\right\} + \eta_\delta\left(\frac{q}{p} - \frac{1-q}{1-p}\right) + \left(\frac{q}{p} - \frac{1-q}{1-p}\right)^2\mathcal{O}\left(\frac{1}{n_{\mathrm{IS}}^2} + \eta_\delta^2\right)
$$

$$
= \frac{q}{n_{\mathrm{IS}}^2}\left(\frac{p}{q} + \frac{1-p}{1-q}\right) + \mathcal{O}\left(\left[\left(\frac{q}{p} - \frac{1-q}{1-p}\right) + \left(\frac{q}{p} - \frac{1-q}{1-p}\right)^2\right]\sqrt{\frac{6p\log(2n_{\mathrm{IS}})}{n_{\mathrm{IS}}}}\right).
$$

This concludes the proof. $\qquad\square$

*Proof of Theorem 1.* Assume a party estimates the Bernoulli distributions $Q_j$ with parameters $q_j$ held by parties $j \in [n]$. The estimating party shares with each of the other parties a common prior $P_j$ in the form of a Bernoulli distribution with parameter $p_j$ and access to unlimited shared randomness.

To help estimate $Q_j$, the $j$-th party sends $K$ samples to the estimator through importance sampling. Therefore, both parties sample $K n_{\mathrm{IS}}$ i.i.d. samples $X_{\ell,i} \sim P_j$ for $\ell \in [K], i \in [n_{\mathrm{IS}}]$, independently and identically from $P_j$. The party holding $Q_j$ constructs for each $\ell \in [K]$ an auxiliary distribution

$$W_\ell(i) = \frac{Q_j(X_{\ell,i})/P_j(X_{\ell,i})}{\sum_{i=1}^{n_{\mathrm{IS}}} Q_j(X_{\ell,i})/P_j(X_{\ell,i})},$$

from which it samples to obtain an index $I_\ell$. The index is transmitted to the estimating party, which reconstructs the corresponding sample $X_{\ell,I_\ell}$. Averaging the samples for all $\ell \in [K]$ gives an estimate $\hat{q}_j$ of $q_j$, i.e., $\hat{q}_j = \frac{1}{K} \sum_{\ell=1}^K X_{\ell,I_\ell}$. This process is repeated for all $j \in [n]$.

We assume that $|q_j - p_j| \leq \rho$ for all $i, j \in [n]$, and that the difference between the priors, is bounded as $|p_i - p_j| \leq \zeta$ for all $i, j \in [n]$. The goal is to bound $\mathrm{d}_{\mathrm{KL}}\left(\frac{1}{n}\sum_{j=1}^n \hat{q}_j \| p_i\right)$ from above for any $i \in [n]$.

By the convexity of KL-divergence, we have

$$\mathrm{d}_{\mathrm{KL}}\left(\frac{1}{n}\sum_{j=1}^n \hat{q}_j \| p_i\right) \leq \frac{1}{n}\sum_{i=1}^n \mathrm{d}_{\mathrm{KL}}\left(\hat{q}_j \| p_i\right).$$

To bound $\mathrm{d}_{\mathrm{KL}}\left(\hat{q}_j \| p_i\right)$ for any $i, j \in [n]$, by the triangle inequality, we can write

$$|\hat{q}_j - p_i| \leq |\hat{q}_j - \Pr(X_\ell = 1)| + |\Pr(X_\ell = 1) - q_j| + |q_j - p_j| + |p_j - p_i|,$$

where $|\hat{q}_j - \Pr(X_\ell = 1)|$ is bounded by Lemma 1. By Hoeffding's inequality, we have with probability at least $1 - \delta'$ that

$$|\hat{q} - \Pr(X_\ell = 1)| \leq \sqrt{\frac{-\ln(\delta'/2)}{2n_{\mathrm{IS}}}}.$$

Thus, with probability at least $1 - \delta'$, since $p_j \leq p_i + \zeta$, we have with $\Delta := \frac{q_j}{p_j - \zeta} - \frac{1 - q_j}{1 - p_j + \zeta}$ and $\Delta'_j := q_j\left(\frac{p_j + \zeta}{q_j} + \frac{1 - p_j + \zeta}{1 - q_j}\right)$ that

$$|\hat{q}_j - p_i| \leq \frac{\Delta'_j}{n_{\mathrm{IS}}^2} + \mathcal{O}\left((\Delta_j + \Delta_j^2)\sqrt{\frac{6(p_i + \zeta)\log(2n_{\mathrm{IS}})}{n_{\mathrm{IS}}}}\right) + \sqrt{\frac{-\ln(\delta'/2)}{2n_{\mathrm{IS}}}} + \rho + \zeta.$$

This holds under the assumption that $p_j > \zeta$ for all $j \in [n]$. By the reversed Pinsker's inequality, we obtain

$$\mathrm{D}_{\mathrm{KL}}\left(\hat{q}_j \| p_i\right) \leq \frac{2}{\min\{p_i, 1 - p_i\}}\left(\frac{\Delta'_j}{n_{\mathrm{IS}}^2} + \mathcal{O}\left((\Delta_j + \Delta_j^2)\sqrt{\frac{6(p_i + \zeta)\log(2n_{\mathrm{IS}})}{n_{\mathrm{IS}}}}\right)\right.$$

$$\left. + \sqrt{\frac{-\ln(\delta'/2)}{2n_{\mathrm{IS}}}} + \rho + \zeta\right)^2.$$

The statement of the theorem follows by the convexity of KL-divergence. □

## B GRADIENT DESCENT WITH A KL-PROXIMITY

Mirror descent employs point-wise optimization in the form of a first-order approximation of $F(\hat{\theta}_t, \mathcal{D}_i)$ with proximity term $D_F(p, q)$, where $D_F$ is the Bregman divergence associated with function $F(\cdot)$. When $F(x) = \|x\|^2$, and hence the Bregman divergence is the Euclidean distance, this is known as gradient descent. Let now $p$ and $q$ be vectors with the entries corresponding to independent Bernoulli parameters. When we choose $F(x) = x\log(x) + (1 - x)\log(1 - x)$, the Bregman divergence becomes $D_F(p, q) = \sum_{k=1}^d \mathrm{D}_{\mathrm{KL}}\left(p_k \| q_k\right)$. Hence, we are optimizing with respect to a KL-proximity constraint. The mapping between dual and primal spaces is then given by $\nabla F(x) = \log(x) - \log(1 - x)$ and $(\nabla F(x))^{-1} = \frac{1}{e^{-x}+1}$, respectively; also known as the inverse sigmoid and the sigmoid functions.

## C  BLOCK ALLOCATION

The simplest yet effective strategy for block allocation is to partition the model into equally-sized blocks of size $d/B$ for importance sampling (Fixed). The partitioning into blocks is required to make importance sampling practically feasible in this setting. It is known that for vanishing importance sampling error, the number of samples $n_{\text{IS}}$ from a block $p_{i,u,b}^t$ of the prior is supposed to be in the order of $\exp\left(\text{D}_{\text{KL}}\left(q_{i,b}^t\|p_{i,u,b}^t\right)\right)$, where $q_{i,b}^t$ is the $b$-th block of posterior $q_i^t$. It was observed by Isik et al. (2024) that the KL-divergence decreases as the training progresses with the global model used as a prior, which is intuitive since the local training will change the posterior less and less as training converges. To adapt the block size according to the divergence from the posterior with respect to the prior, Isik et al. (2024) proposed an adaptive block allocation strategy (Adaptive), where upon realizing a large deviation from the target KL-divergence per block, clients partition their model into blocks with equal sums of parameter-wise KL-divergences and transmit the block intervals to the federator. The federator aggregates the indices of all the clients, and broadcasts the updated block allocation. We propose in this work a low complexity solution that adapts the block size according to the average KL-divergence per block (Adaptive-Avg). This alleviates the cost of computing and transmitting the exact block partitions, where the transmission of each block size requires $\log_2(b_{\max})$ bits, with $b_{\max}$ the maximum pre-defined block size. Instead, the transmission of one size is enough in our solution. If the average KL per block $\text{D}_{\text{KL}}\left(q_{i,b}^t\|p_{i,u,b}^t\right)$ deviates more than a given factor, the clients request to update the blocks. In the next iteration, each client proposes a block size, and the federator averages and broadcasts an updated size.

## D  ADDITIONAL EXPERIMENTAL DETAILS

We use the cross-entropy loss and a batch size of 128 in all our experiments. We use Adam (Kingma & Ba, 2015) as an optimizer with learning rate $\eta = 0.0003$ for all non-stochastic methods, and $\eta = 0.1$ for probabilistic mask training. For non-stochastic FL, we use a federator (server) learning rate of 0.1, i.e., the clients' gradients are averaged, and the federator updates the global model with learning rate 0.1. Solely for M3, we use a federator learning rate of 0.02 to obtain reliable results. For LIEC and CSER, we use an average period of 50 global iterations (cf. (Cheng et al., 2024; Xie et al., 2020)). For M3, we use TopK with $K = \lfloor d/n \rfloor$. To run the simulations, we use a cluster of different architectures, which we list in the following table.

| CPU(s) | RAM | GPU(s) | VRAM |
|---|---|---|---|
| 2x Intel Xeon Platinum 8176 (56 cores) | 256 GB | 2x NVIDIA GeForce GTX 1080 Ti | 11 GB |
| 2x AMD EPYC 7282 (32 cores) | 512 GB | NVIDIA GeForce RTX 4090 | 24 GB |
| 2x AMD EPYC 7282 (32 cores) | 640 GB | NVIDIA GeForce RTX 4090 | 24 GB |
| 2x AMD EPYC 7282 (32 cores) | 448 GB | NVIDIA GeForce RTX 4080 | 16 GB |
| 2x AMD EPYC 7282 (32 cores) | 256 GB | NVIDIA GeForce RTX 4080 | 16 GB |
| HGX-A100 (96 cores) | 1 TB | 4x NVIDIA A100 | 80 GB |
| DGX-A100 (252 cores) | 2 TB | 8x NVIDIA Tesla A100 | 80 GB |
| DGX-1-V100 (76 cores) | 512 GB | 8x NVIDIA Tesla V100 | 16 GB |
| DGX-1-P100 (76 cores) | 512 GB | 8x NVIDIA Tesla P100 | 16 GB |
| HPE-P100 (28 cores) | 256 GB | 4x NVIDIA Tesla P100 | 16 GB |

Table 1: System specifications of our simulation cluster.

The details of the CNN architectures used in our experiments are summarized in the following. The parameter count is 61706 for LeNet5, 1933258 for 4CNN, and 2262602 for 6CNN.

Table 2: LeNet5 Architecture Overview

| Layer | Specification | Activation |
|-------|---------------|------------|
| 5x5 Conv | 6 filters, stride 1 | ReLU, AvgPool (2x2) |
| 5x5 Conv | 16 filters, stride 1 | ReLU, AvgPool (2x2) |
| Linear | 120 units | ReLU |
| Linear | 84 units | ReLU |
| Linear | 10 units | Softmax |

Table 3: 4-layer CNN (4CNN) Architecture Overview

| Layer | Specification | Activation |
|-------|---------------|------------|
| 3x3 Conv | 64 filters, stride 1 | ReLU |
| 3x3 Conv | 64 filters, stride 1 | ReLU, MaxPool (2x2) |
| 3x3 Conv | 128 filters, stride 1 | ReLU |
| 3x3 Conv | 128 filters, stride 1 | ReLU, MaxPool (2x2) |
| Linear | 256 units | ReLU |
| Linear | 256 units | ReLU |
| Linear | 10 units | Softmax |

Table 4: 6-layer CNN (6CNN) Architecture Overview

| Layer | Specification | Activation |
|-------|---------------|------------|
| 3x3 Conv | 64 filters, stride 1 | ReLU |
| 3x3 Conv | 64 filters, stride 1 | ReLU, MaxPool (2x2) |
| 3x3 Conv | 128 filters, stride 1 | ReLU |
| 3x3 Conv | 128 filters, stride 1 | ReLU, MaxPool (2x2) |
| 3x3 Conv | 256 filters, stride 1 | ReLU |
| 3x3 Conv | 256 filters, stride 1 | ReLU, MaxPool (2x2) |
| Linear | 256 units | ReLU |
| Linear | 256 units | ReLU |
| Linear | 10 units | Softmax |

For the sake of clarity, in the paper we restrict the analysis to a fixed number of importance samples $n_{\mathrm{IS}}$, block sizes $B$, and choice of priors $p_{i,u}^t, p_{i,d}^t$. Our experiments have shown that, while increasing $n_{\mathrm{IS}}$ beyond the ones used in our algorithms slightly improves the convergence over the number of epochs, the convergence with respect to the communication cost did not significantly improve. The block size is mainly limited by the system resources at hand, and one would choose the largest possible for best efficiency while complying with memory resources. We investigated many different prior choices and found the former global model to be reasonably good in almost all cases. With high heterogeneity, it might be beneficial to use different convex combinations as priors, which mix the former global model with the latest posterior estimate of a certain client, but the gains we experienced were minor. Hence, we settled on the former global estimate for simplicity in presenting the algorithm.

## E ADDITIONAL EXPERIMENTS

We provide in the following experiments for both uniform (i.i.d.) and heterogeneous (non-i.i.d.) data distributions for training LeNet5 and a 4-layer CNN on MNIST, a 4-layer CNN on Fashion MNIST, and a 6-layer CNN on CIFAR-10. The details of the neural networks can be found in Tables 2 to 4. For each setting and method depicted, we show the average of three simulation runs with different seeds. We plot for each setting the test accuracies over the communication cost in bits, and the maximum test accuracy over the bitrate. We provide tables summarizing the maximum test accuracies with their standard deviation over multiple runs, the total bitrates and the bitrates split into uplink and downlink. The overall bitrates per parameter (bpp) are computed assuming

point-to-point links between all participants, i.e., uplink and downlink costs have equal weight. For the case when a broadcast (BC) link between the federator and the clients is available, the bitrate per parameter for all baseline schemes reduces by a factor of $n$. BICOMPFL-GRprofits similarly from the broadcast link, but BICOMPFL-PRcannot profit due to the absence of shared randomness, giving the same overall bitrate compared to the point-to-point link scenario. We highlight for each of the measures the scheme with the best result. Consistently throughout all experiments, BICOMPFL achieves order-wise savings in the bitrates per parameter while reaching state-of-the-art accuracies in the classification task. While the sampling can introduce an additional computational overhead depending on the implementation, the storage cost is similar to the baselines. Since we leverage as priors the former global model, the additional storage cost incurred is limited to storing until the next iteration the estimate of the former global model at each client, i.e., where the training started, which is usually not a bottleneck. This can be cheaper than some baselines, which require storing data for momentum and error-feedback.

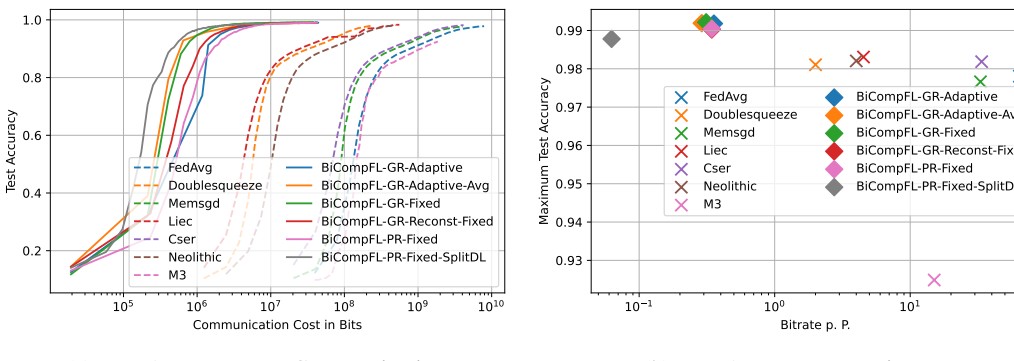

(a) Test Accuracy over Communication       (b) Test Accuracy over Bitrate

Figure 3: MNIST LeNet i.i.d.

For LeNet5 on MNIST, it can be observed that all our proposed methods converge significantly faster to satisfying accuracies with respect to the communication cost, while achieving higher maximum accuracies after 200 epochs than the non-stochastic baselines. Partitioning the model on the downlink can help to further reduce the communication cost with only a minor loss in performance, especially in the i.i.d. setting. For non-i.i.d. data distribution, the loss in performance is larger than for i.i.d. distribution. However, at the beginning of the training, the model improves faster with respect to the communication cost than all other schemes. The bitrates are comparable for all our methods, with the exception of BICOMPFL-PR-Fixed-SplitDL. Further, BICOMPFL-GR-Reconst-Fixed does not suffer notable performance degradation from employing an additional importance sampling step (especially for i.i.d. data allocation).

Table 5: MNIST LeNet i.i.d.

| Method | Acc (mean ± std) | bpp | bpp (BC) | Uplink | Downlink |
|---|---|---|---|---|---|
| FedAvg | $0.978 \pm 0.1$ | 64.0 | 35.0 | 32.0 | 32.0 |
| Doublesqueeze | $0.981 \pm 0.1$ | 2.0 | 1.1 | 1.0 | 1.0 |
| Memsgd | $0.977 \pm 0.1$ | 33.0 | 4.2 | 1.0 | 32.0 |
| Liec | $0.983 \pm 0.1$ | 4.5 | 2.5 | 2.3 | 2.3 |
| Cser | $0.982 \pm 0.09$ | 34.0 | 4.3 | 1.0 | 33.0 |
| Neolithic | $0.982 \pm 0.1$ | 4.0 | 2.2 | 2.0 | 2.0 |
| M3 | $0.925 \pm 0.2$ | 15.0 | 2.2 | 8.0 | 7.1 |
| BiCompFL-GR-Adaptive | $\mathbf{0.992 \pm 0.0006}$ | 0.36 | 0.068 | 0.036 | 0.32 |
| BiCompFL-GR-Adaptive-Avg | $0.992 \pm 0.0003$ | 0.29 | **0.055** | **0.029** | 0.26 |
| BiCompFL-GR-Fixed | $0.992 \pm 0.0002$ | 0.31 | 0.059 | 0.031 | 0.28 |
| BiCompFL-GR-Reconst-Fixed | $0.99 \pm 0.0002$ | 0.34 | 0.063 | 0.031 | 0.31 |
| BiCompFL-PR-Fixed | $0.99 \pm 0.0004$ | 0.34 | 0.34 | 0.031 | 0.31 |
| BiCompFL-PR-Fixed-SplitDL | $0.988 \pm 0.0009$ | **0.063** | 0.063 | 0.031 | **0.031** |

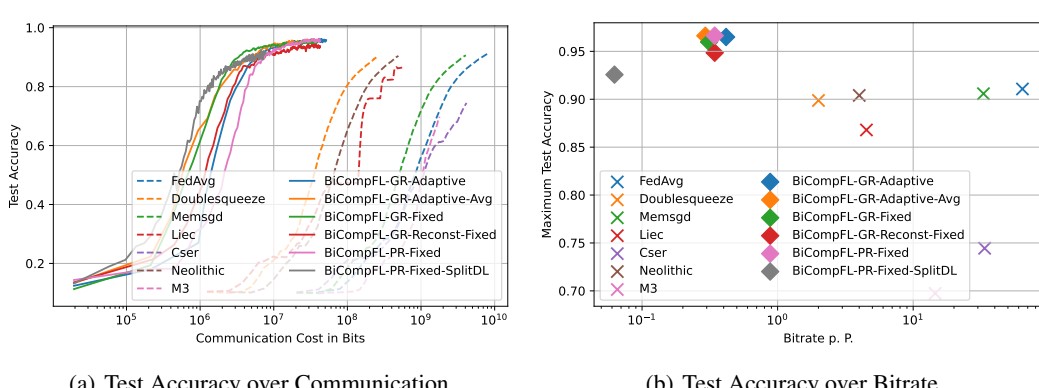

(a) Test Accuracy over Communication

(b) Test Accuracy over Bitrate

Figure 4: MNIST LeNet non-i.i.d.

Table 6: MNIST LeNet non-i.i.d.

| Method | Acc (mean ± std) | bpp | bpp (BC) | Uplink | Downlink |
|---|---|---|---|---|---|
| FedAvg | 0.911 ± 0.2 | 64.0 | 35.0 | 32.0 | 32.0 |
| Doublesqueeze | 0.899 ± 0.2 | 2.0 | 1.1 | 1.0 | 1.0 |
| Memsgd | 0.906 ± 0.2 | 33.0 | 4.2 | 1.0 | 32.0 |
| Liec | 0.866 ± 0.2 | 4.5 | 2.5 | 2.3 | 2.3 |
| Cser | 0.744 ± 0.2 | 34.0 | 4.3 | 1.0 | 33.0 |
| Neolithic | 0.904 ± 0.2 | 4.0 | 2.2 | 2.0 | 2.0 |
| M3 | 0.697 ± 0.2 | 15.0 | 2.2 | 7.3 | 7.2 |
| BiCompFL-GR-Adaptive | 0.965 ± 0.02 | 0.42 | 0.079 | 0.042 | 0.37 |
| BiCompFL-GR-Adaptive-Avg | **0.966 ± 0.02** | 0.29 | **0.056** | **0.029** | 0.26 |
| BiCompFL-GR-Fixed | 0.96 ± 0.03 | 0.31 | 0.059 | 0.031 | 0.28 |
| BiCompFL-GR-Reconst-Fixed | 0.949 ± 0.03 | 0.34 | 0.063 | 0.031 | 0.31 |
| BiCompFL-PR-Fixed | 0.966 ± 0.02 | 0.34 | 0.34 | 0.031 | 0.31 |
| BiCompFL-PR-Fixed-SplitDL | 0.926 ± 0.04 | **0.063** | 0.063 | 0.031 | **0.031** |

For 4CNN trained on MNIST, the differences between the proposed approaches become more visible. In the i.i.d. setting, we can observe that the adaptive block allocations (both Adaptive and Adaptive-Avg) can drastically reduce the average bitrate in BICOMPFL-GR. Partitioning the model in the downlink (BICOMPFL-PR-Fixed-SplitDL) improves the accuracy over bitrate significantly compared to BICOMPFL-PR-Fixed.

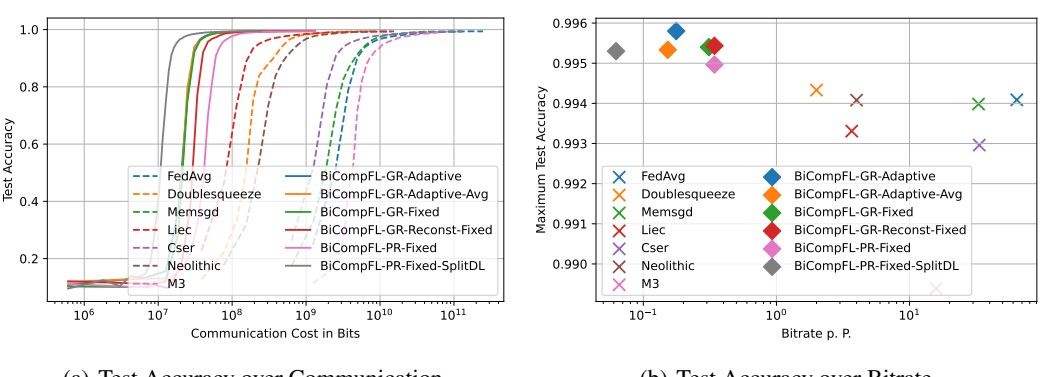

(a) Test Accuracy over Communication

(b) Test Accuracy over Bitrate

Figure 5: MNIST 4CNN i.i.d.

Table 7: MNIST 4CNN i.i.d.

| Method | Acc (mean ± std) | bpp | bpp (BC) | Uplink | Downlink |
|---|---|---|---|---|---|
| FedAvg | $0.994 \pm 0.06$ | 64.0 | 35.0 | 32.0 | 32.0 |
| Doublesqueeze | $0.994 \pm 0.1$ | 2.0 | 1.1 | 1.0 | 1.0 |
| Memsgd | $0.994 \pm 0.08$ | 33.0 | 4.2 | 1.0 | 32.0 |
| Liec | $0.993 \pm 0.07$ | 3.7 | 2.0 | 1.8 | 1.8 |
| Cser | $0.993 \pm 0.06$ | 33.0 | 4.3 | 1.0 | 32.0 |
| Neolithic | $0.994 \pm 0.08$ | 4.0 | 2.2 | 2.0 | 2.0 |
| M3 | $0.989 \pm 0.2$ | 16.0 | 2.2 | 8.4 | 7.4 |
| BiCompFL-GR-Adaptive | $\mathbf{0.996 \pm 0.0001}$ | 0.18 | 0.034 | 0.018 | 0.16 |
| BiCompFL-GR-Adaptive-Avg | $0.995 \pm 0.0001$ | 0.15 | **0.029** | **0.015** | 0.14 |
| BiCompFL-GR-Fixed | $0.995 \pm 0.0002$ | 0.31 | 0.059 | 0.031 | 0.28 |
| BiCompFL-GR-Reconst-Fixed | $0.995 \pm 0.0001$ | 0.34 | 0.062 | 0.031 | 0.31 |
| BiCompFL-PR-Fixed | $0.995 \pm 0.0002$ | 0.34 | 0.34 | 0.031 | 0.31 |
| BiCompFL-PR-Fixed-SplitDL | $0.995 \pm 0.0002$ | **0.062** | 0.062 | 0.031 | **0.031** |

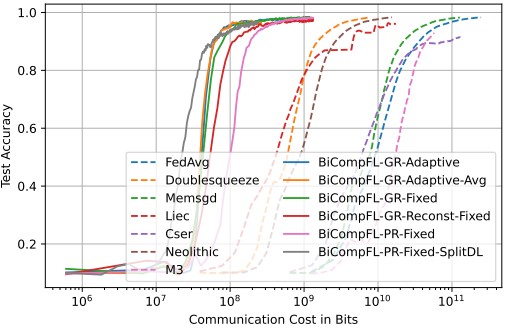

(a) Test Accuracy over Communication

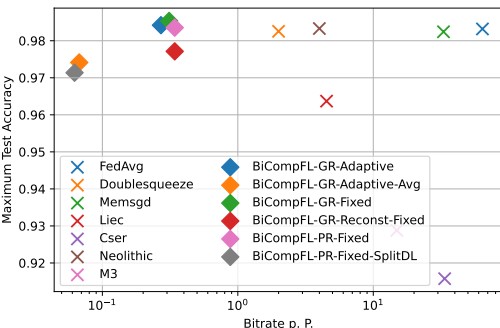

(b) Test Accuracy over Bitrate

Figure 6: MNIST 4CNN non-i.i.d.

In the non-i.i.d. case of 4CNN on MNIST, the adaptive average allocation strategy provides a significant reduction in the bitrate for BICOMPFL-GR, with similar loss in the accuracy as SplitDL for BICOMPFL-PR. In this setting, it is also apparent that the reconstruction in BICOMPFL-GR degrades the performance without gains in the bitrate compared to the proposed Algorithm 2.

Table 8: MNIST 4CNN non-i.i.d.

| Method | Acc (mean ± std) | bpp | bpp (BC) | Uplink | Downlink |
|---|---|---|---|---|---|
| FedAvg | $0.983 \pm 0.1$ | 64.0 | 35.0 | 32.0 | 32.0 |
| Doublesqueeze | $0.982 \pm 0.2$ | 2.0 | 1.1 | 1.0 | 1.0 |
| Memsgd | $0.982 \pm 0.2$ | 33.0 | 4.2 | 1.0 | 32.0 |
| Liec | $0.963 \pm 0.2$ | 4.5 | 2.5 | 2.3 | 2.3 |
| Cser | $0.915 \pm 0.1$ | 34.0 | 4.3 | 1.0 | 33.0 |
| Neolithic | $0.983 \pm 0.2$ | 4.0 | 2.2 | 2.0 | 2.0 |
| M3 | $0.929 \pm 0.3$ | 15.0 | 2.2 | 7.8 | 7.1 |
| BiCompFL-GR-Adaptive | $0.984 \pm 0.009$ | 0.27 | 0.051 | 0.026 | 0.24 |
| BiCompFL-GR-Adaptive-Avg | $0.974 \pm 0.02$ | 0.067 | **0.013** | **0.0068** | 0.061 |
| BiCompFL-GR-Fixed | $\mathbf{0.985 \pm 0.008}$ | 0.31 | 0.059 | 0.031 | 0.28 |
| BiCompFL-GR-Reconst-Fixed | $0.977 \pm 0.01$ | 0.34 | 0.062 | 0.031 | 0.31 |
| BiCompFL-PR-Fixed | $0.984 \pm 0.009$ | 0.34 | 0.34 | 0.031 | 0.31 |
| BiCompFL-PR-Fixed-SplitDL | $0.971 \pm 0.02$ | **0.062** | 0.062 | 0.031 | **0.031** |

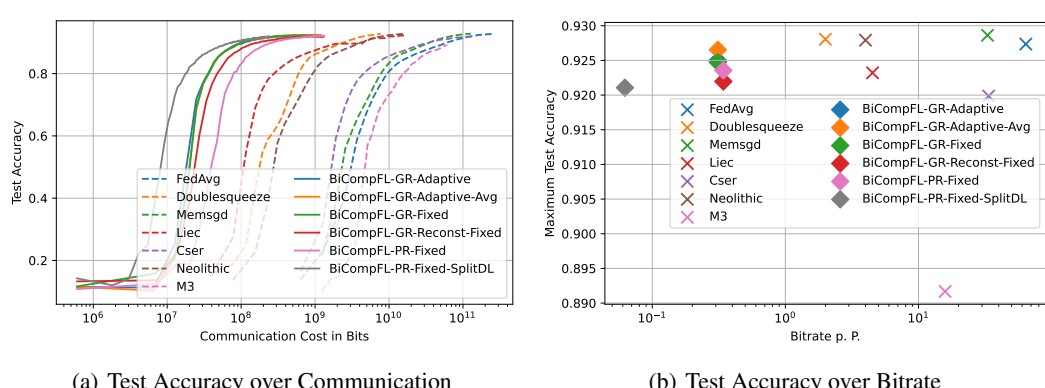

(a) Test Accuracy over Communication
(b) Test Accuracy over Bitrate

Figure 7: Fashion MNIST 4CNN i.i.d.

Table 9: Fashion MNIST 4CNN i.i.d.

| Method | Acc (mean ± std) | bpp | bpp (BC) | Uplink | Downlink |
|---|---|---|---|---|---|
| FedAvg | $0.927 \pm 0.07$ | 64.0 | 35.0 | 32.0 | 32.0 |
| Doublesqueeze | $\mathbf{0.928 \pm 0.1}$ | 2.0 | 1.1 | 1.0 | 1.0 |
| Memsgd | $0.928 \pm 0.09$ | 33.0 | 4.2 | 1.0 | 32.0 |
| Liec | $0.923 \pm 0.08$ | 4.5 | 2.5 | 2.3 | 2.3 |
| Cser | $0.92 \pm 0.08$ | 34.0 | 4.3 | 1.0 | 33.0 |
| Neolithic | $0.928 \pm 0.09$ | 4.0 | 2.2 | 2.0 | 2.0 |
| M3 | $0.892 \pm 0.2$ | 16.0 | 2.2 | 8.3 | 7.6 |
| BiCompFL-GR-Adaptive | $0.925 \pm 0.001$ | 0.31 | $\mathbf{0.059}$ | $\mathbf{0.031}$ | 0.28 |
| BiCompFL-GR-Adaptive-Avg | $0.927 \pm 0.0007$ | 0.31 | $\mathbf{0.059}$ | $\mathbf{0.031}$ | 0.28 |
| BiCompFL-GR-Fixed | $0.925 \pm 0.0007$ | 0.31 | $\mathbf{0.059}$ | $\mathbf{0.031}$ | 0.28 |
| BiCompFL-GR-Reconst-Fixed | $0.922 \pm 0.001$ | 0.34 | 0.062 | $\mathbf{0.031}$ | 0.31 |
| BiCompFL-PR-Fixed | $0.924 \pm 0.002$ | 0.34 | 0.34 | $\mathbf{0.031}$ | 0.31 |
| BiCompFL-PR-Fixed-SplitDL | $0.921 \pm 0.002$ | $\mathbf{0.062}$ | 0.062 | $\mathbf{0.031}$ | $\mathbf{0.031}$ |

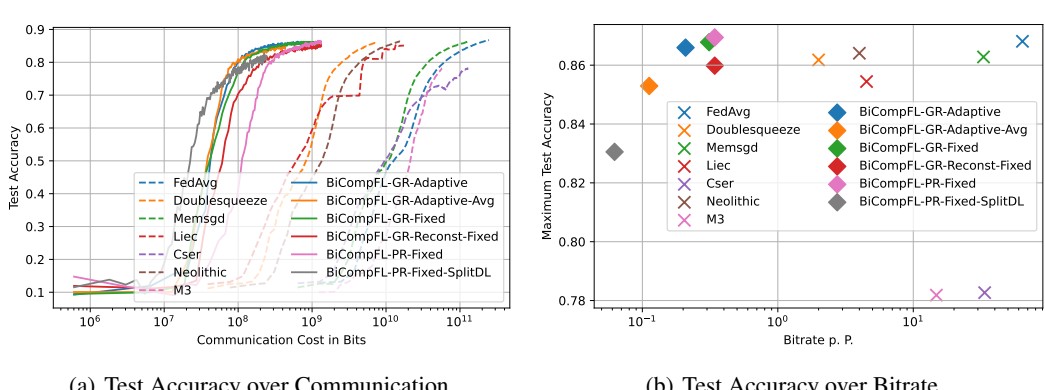

(a) Test Accuracy over Communication
(b) Test Accuracy over Bitrate

Figure 8: Fashion MNIST 4CNN non-i.i.d.

The results for Fashion MNIST are similar compared to the MNIST case. However, it becomes clear that BICOMPFL-PR can significantly suffer from the unavailability of shared randomness in terms of the achieved accuracy when data is highly heterogeneous.

Table 10: Fashion MNIST 4CNN non-i.i.d.

| Method | Acc (mean ± std) | bpp | bpp (BC) | Uplink | Downlink |
|---|---|---|---|---|---|
| FedAvg | 0.867 ± 0.1 | 64.0 | 35.0 | 32.0 | 32.0 |
| Doublesqueeze | 0.861 ± 0.2 | 2.0 | 1.1 | 1.0 | 1.0 |
| Memsgd | 0.863 ± 0.2 | 33.0 | 4.2 | 1.0 | 32.0 |
| Liec | 0.853 ± 0.1 | 4.5 | 2.5 | 2.3 | 2.3 |
| Cser | 0.781 ± 0.1 | 34.0 | 4.3 | 1.0 | 33.0 |
| Neolithic | 0.864 ± 0.2 | 4.0 | 2.2 | 2.0 | 2.0 |
| M3 | 0.782 ± 0.2 | 15.0 | 2.2 | 8.0 | 6.9 |
| BiCompFL-GR-Adaptive | 0.866 ± 0.03 | 0.21 | 0.04 | 0.021 | 0.19 |
| BiCompFL-GR-Adaptive-Avg | 0.853 ± 0.04 | 0.11 | **0.021** | **0.011** | 0.1 |
| BiCompFL-GR-Fixed | 0.868 ± 0.03 | 0.31 | 0.059 | 0.031 | 0.28 |
| BiCompFL-GR-Reconst-Fixed | 0.86 ± 0.02 | 0.34 | 0.062 | 0.031 | 0.31 |
| BiCompFL-PR-Fixed | **0.869 ± 0.03** | 0.34 | 0.34 | 0.031 | 0.31 |
| BiCompFL-PR-Fixed-SplitDL | 0.831 ± 0.03 | **0.062** | 0.062 | 0.031 | **0.031** |

For 6CNN trained on CIFAR-10, the negative effects of missing global shared randomness and reconstructing in the case of BICOMPFL-GR are prominent. For non-i.i.d. data distributions, the adaptive average allocation shows improvements over the fixed or the average block allocation. Partitioning the model is not a viable option in this setting, especially under non-i.i.d. data.

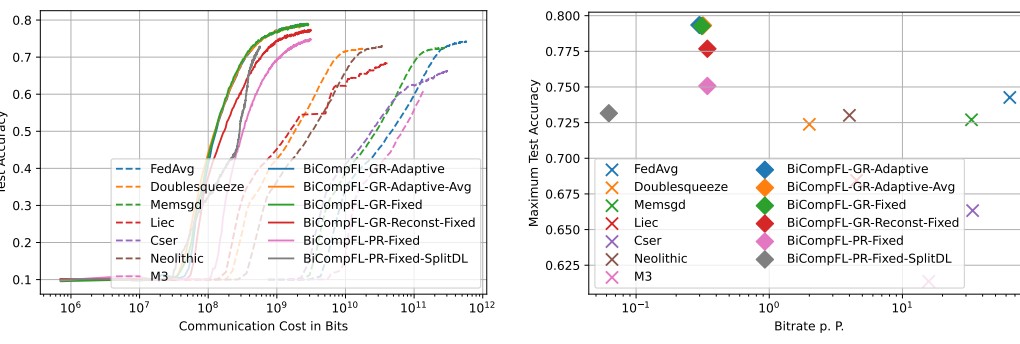

(a) Test Accuracy over Communication            (b) Test Accuracy over Bitrate

Figure 9: CIFAR-10 6CNN i.i.d.

Table 11: CIFAR-10 6CNN i.i.d.

| Method | Acc (mean ± std) | bpp | bpp (BC) | Uplink | Downlink |
|---|---|---|---|---|---|
| FedAvg | 0.742 ± 0.1 | 64.0 | 35.0 | 32.0 | 32.0 |
| Doublesqueeze | 0.723 ± 0.1 | 2.0 | 1.1 | 1.0 | 1.0 |
| Memsgd | 0.727 ± 0.1 | 33.0 | 4.2 | 1.0 | 32.0 |
| Liec | 0.684 ± 0.09 | 4.5 | 2.5 | 2.3 | 2.3 |
| Cser | 0.663 ± 0.08 | 34.0 | 4.3 | 1.0 | 33.0 |
| Neolithic | 0.73 ± 0.1 | 4.0 | 2.2 | 2.0 | 2.0 |
| M3 | 0.614 ± 0.1 | 16.0 | 2.2 | 8.3 | 7.5 |
| BiCompFL-GR-Adaptive | **0.793 ± 0.002** | 0.3 | **0.057** | **0.03** | 0.27 |
| BiCompFL-GR-Adaptive-Avg | 0.793 ± 0.002 | 0.32 | 0.061 | 0.032 | 0.29 |
| BiCompFL-GR-Fixed | 0.793 ± 0.004 | 0.31 | 0.059 | 0.031 | 0.28 |
| BiCompFL-GR-Reconst-Fixed | 0.777 ± 0.002 | 0.34 | 0.062 | 0.031 | 0.31 |
| BiCompFL-PR-Fixed | 0.751 ± 0.003 | 0.34 | 0.34 | 0.031 | 0.31 |
| BiCompFL-PR-Fixed-SplitDL | 0.732 ± 0.02 | **0.062** | 0.062 | 0.031 | **0.031** |

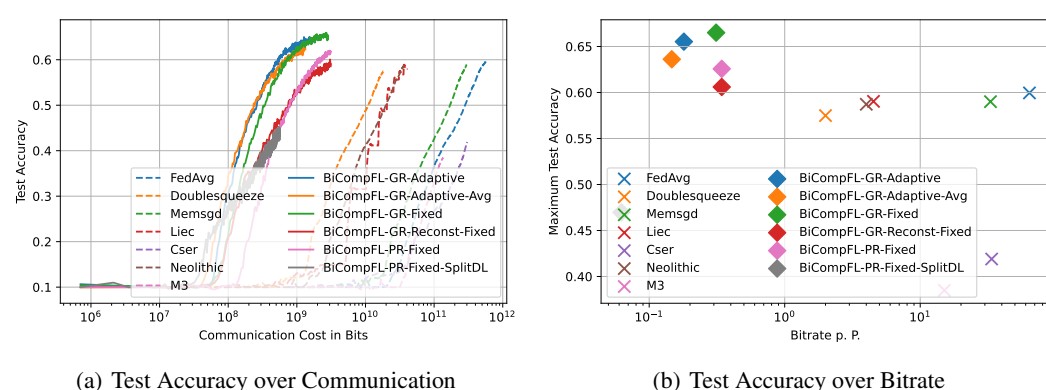

(a) Test Accuracy over Communication

(b) Test Accuracy over Bitrate

Figure 10: CIFAR-10 6CNN non-i.i.d.

Table 12: CIFAR-10 6CNN non-i.i.d.

| Method | Acc (mean ± std) | bpp | bpp (BC) | Uplink | Downlink |
|---|---|---|---|---|---|
| FedAvg | $0.599 \pm 0.1$ | 64.0 | 35.0 | 32.0 | 32.0 |
| Doublesqueeze | $0.575 \pm 0.1$ | 2.0 | 1.1 | 1.0 | 1.0 |
| Memsgd | $0.589 \pm 0.1$ | 33.0 | 4.2 | 1.0 | 32.0 |
| Liec | $0.589 \pm 0.2$ | 4.5 | 2.5 | 2.3 | 2.3 |
| Cser | $0.419 \pm 0.09$ | 34.0 | 4.3 | 1.0 | 33.0 |
| Neolithic | $0.587 \pm 0.1$ | 4.0 | 2.2 | 2.0 | 2.0 |
| M3 | $0.385 \pm 0.1$ | 15.0 | 2.2 | 8.3 | 6.7 |
| BiCompFL-GR-Adaptive | $0.655 \pm 0.04$ | 0.18 | 0.034 | 0.018 | 0.16 |
| BiCompFL-GR-Adaptive-Avg | $0.636 \pm 0.05$ | 0.15 | **0.028** | **0.015** | 0.13 |
| BiCompFL-GR-Fixed | **$0.665 \pm 0.03$** | 0.31 | 0.059 | 0.031 | 0.28 |
| BiCompFL-GR-Reconst-Fixed | $0.606 \pm 0.05$ | 0.34 | 0.062 | 0.031 | 0.31 |
| BiCompFL-PR-Fixed | $0.626 \pm 0.03$ | 0.34 | 0.34 | 0.031 | 0.31 |
| BiCompFL-PR-Fixed-SplitDL | $0.47 \pm 0.07$ | **0.062** | 0.062 | 0.031 | **0.031** |

For completeness, we present in what follows the test accuracies over the number of trained epochs for all scenarios considered above. The setting of interest to this work is that of limited communication cost, and in particular, which performance is achievable given a fixed communication budget. Nonetheless, we can find that our proposed methods are not inferior in convergence speed over epochs compared to the baselines.

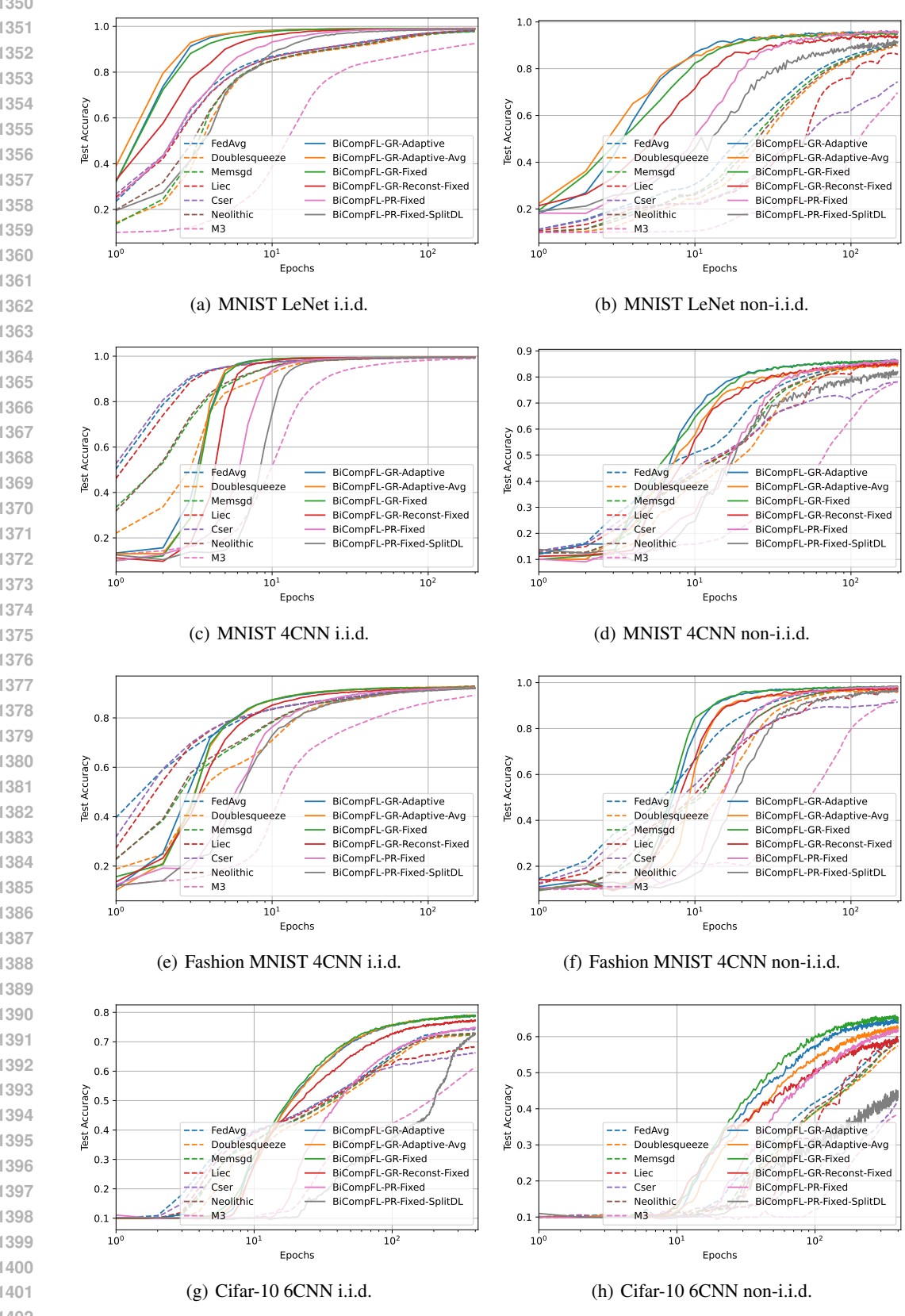

Figure 11: Test Accuracy over Epochs

