# OpenReview forum: "BiCompFL: Stochastic Federated Learning with Bi-Directional Compression"
_ICLR.cc/2025/Conference — Submitted to ICLR 2025_

### Official Review · Reviewer_on4o · 2024-10-28

**Soundness:** 3
**Presentation:** 3
**Contribution:** 2
**Rating:** 5
**Confidence:** 4

**Summary:**

This paper addresses a critical challenge in Federated Learning (FL) — the communication bottleneck that arises due to the extensive data exchange between clients and the federator. Specifically, the authors introduce **BICOMPFL**, a novel federated learning framework aimed at reducing communication costs in both uplink (clients to federator) and downlink (federator to clients) transmissions. BICOMPFL employs bi-directional stochastic compression by leveraging importance sampling and utilizing carefully chosen prior distributions as side information to transmit model updates efficiently. The framework presents two variants: **BICOMPFL-GR (Global Shared Randomness)**, which assumes a globally shared randomness source enabling synchronized sampling across all clients, and **BICOMPFL-PR (Private Shared Randomness)**, which operates with only private shared randomness between each client and the federator, addressing scenarios where global shared randomness is infeasible.

**Strengths:**

- While existing research predominantly concentrates on optimizing uplink communication in FL, this paper broadens the scope by simultaneously addressing both uplink and downlink communication bottlenecks.
- Integrating bi-directional stochastic compression through importance sampling, showcases a creative combination of existing techniques tailored to a new problem formulation. Developing two distinct variants—**BICOMPFL-GR (Global Shared Randomness)** and **BICOMPFL-PR (Private Shared Randomness)**—demonstrates originality in catering to different practical scenarios.
- The paper extends and refines existing theories on importance sampling, particularly for Bernoulli distributions.
- The paper provides a theoretical analysis of the proposed framework focusing on an upper bound on communication costs and the interplay between uplink and downlink efficiencies.
- The introduction of adaptive block allocation strategies, including both fixed and adaptive average allocations, demonstrates a thoughtful approach to optimizing model parameter partitioning based on KL-divergence metrics.
- The authors conduct comprehensive numerical experiments across multiple datasets (MNIST, Fashion-MNIST, CIFAR-10) and neural network architectures (LeNet-5, 4CNN, 6CNN), showcasing the versatility and robustness of BICOMPFL.
- The empirical findings consistently demonstrate multi-fold reductions in communication costs compared to state-of-the-art FL methods, without compromising model accuracy.

**Weaknesses:**

- The paper builds upon existing frameworks such as FedPM (Isik et al., 2023) and KLMS (Isik et al., 2024), integrating them into a bi-directional compression strategy. However, the novelty of combining these specific techniques with importance sampling and adaptive block allocation makes the technical novelty somehow limited.
- The theoretical analysis is limited to showing a bound on the cost of downlink transmission. The paper lacks a proper convergence analysis of the proposed algorithm as opposed to [1] and [2].

[1] Constantin Philippenko and Aymeric Dieuleveut. Bidirectional compression in heterogeneous settings for distributed or federated learning with partial participation: tight convergence guarantees. arXiv preprint arXiv:2006.14591, 2020.

[2] Constantin Philippenko and Aymeric Dieuleveut. Preserved central model for faster bidirectional compression in distributed settings. Advances in Neural Information Processing Systems, 34: 2387–2399, 2021.

- The empirical evaluation is confined to well-known datasets (MNIST, Fashion-MNIST, CIFAR-10) and relatively simple CNN architectures (LeNet-5, 4CNN, 6CNN). While these are standard benchmarks, they may not fully demonstrate the framework's effectiveness in more complex or large-scale real-world scenarios.
- While the paper mentions that global shared randomness does not incur additional communication costs, in practice, initializing and maintaining shared randomness (especially as the number of clients scales) may introduce hidden overheads.
- The theoretical results focus on Bernoulli distributions, which may limit the applicability of the analysis to more complex model parameter distributions encountered in practical deep learning models.
- The paper does not extensively explore how sensitive BICOMPFL is to various hyperparameters, such as the number of importance samples (nIS), block sizes (B), or the choice of priors (p).
- The proposed importance sampling and adaptive block allocation strategies may have implications for the energy consumption and computational load on client devices, which is particularly important in edge or mobile FL applications.

**Questions:**

-	Can the theoretical analysis be extended beyond Bernoulli distributions to more complex parameter distributions?
-	Why are the authors not comparing their method to [1] and [2] in the simulation setting? Also, a similar convergence analysis to these references is missing from the paper.

[1] Constantin Philippenko and Aymeric Dieuleveut. Bidirectional compression in heterogeneous settings for distributed or federated learning with partial participation: tight convergence guarantees. arXiv preprint arXiv:2006.14591, 2020.

[2] Constantin Philippenko and Aymeric Dieuleveut. Preserved central model for faster bidirectional compression in distributed settings. Advances in Neural Information Processing Systems, 34: 2387–2399, 2021.

-	The current experiments are limited to MNIST, Fashion-MNIST, and CIFAR-10 with relatively simple CNN architectures. Consider evaluating BICOMPFL on larger and more diverse datasets, such as ImageNet, or involving more complex models like ResNet or Transformers.
-	How feasible is the assumption of shared randomness in large-scale or decentralized FL environments? In practical deployments with numerous clients, maintaining global shared randomness can be challenging.
-	What are the computational and energy overheads associated with importance sampling and adaptive block allocation in BICOMPFL? Especially for clients with limited computational resources, these overheads could impact the feasibility of deployment.
-	How does BICOMPFL perform as the number of clients and the size of the model increase? The current number of clients is somehow very small.
-	How sensitive is BICOMPFL to the choice of hyperparameters, such as the number of importance samples (nIS), block sizes (B), and priors (p)?

---

> ### Author Response · Authors · 2024-11-20
>
> - **S1-S7**: We thank the Reviewer for their positive evaluation of our work.
> - **W1**: Our intention is not to extend FedPM to include downlink (DL) compression, but rather to address DL for general stochastic FL. This is well-motivated since  many works (cf. review in Sec. 6) study the specific caveats of joint uplink (UL) and DL compression in non-stochastic FL, requiring refined analysis and methods. To the best of our knowledge, we are the first to bring this aspect to stochastic FL. There are several  challenges associated with the inclusion of DL compression in the model:
>    1. The role of shared randomness: In UL-only compression, only private shared randomness is required. However, in bi-directional compression, the absence of global shared randomness may lead to divergence of the learning process; in particular when downlink compression is not carefully designed, and clients start their local training from very different models. This aspect is non-critical for deterministic compression methods; and therefore, was not previously explored. However, this aspect is crucial for stochastic FL via importance sampling. Carefully examining the role of different notions of shared randomness is a novel aspect of our paper.
>     2. Analysis and choice of side-information in the case of synchronized and non-synchronized compression, that is, a  distinction between  global shared randomness and private shared randomness. The introduction of additional noise in the training process by individual compression noise on each of the clients strongly affects the learning process and depends crucially on the prior choice. Controlling the noise introduced on the DL as a function of the choice of the prior is thus highly critical. The choice of priors we propose in the paper may seem natural, but is in fact the result of a careful investigation of different priors in different settings of heterogeneity (we only touch upon this in the paper). Conditioned on the prior choice, we analyze the error made by the importance sampling-based compression.
>    3. Analysis of communication cost: We develop a new proof strategy to provide analytical results for the quality of the federator's estimate of the client updates. By contrast, Isik '24 just used generic results from importance sampling, which are inapplicable to the case of bi-directional compression. This result can also be seen as a refined version of the result of Isik '24. Specializing the analysis to Bernoulli parameters enables a tractable  analysis that preserves tightness, which we find more practically relevant than existing theoretical bounds. We believe the proof method is novel, and can be of interest on its own as it can be used for deriving similar results for other types of distributions.
>
>     In general, we would like to highlight that our method applies to any stochastic FL framework. We have adopted FedPM of Isik et al. only for experimental validation. This is motivated by the fact that it achieves high accuracy, but also allows to accurately quantify the communication cost. Given your comment, we understand that the submitted manuscript did not clearly convey the general challenges of bi-directional compression for stochastic FL. We have added a few clarifying sentences in the updated version of the manuscript based on the above discussion, and we believe that this is now clear.
> - **W2**: In general, deriving tight convergence guarantees for stochastic FL algorithms (such as FedPM) is rather challenging. Our particular focus in this work is the communication cost, and so we focused on thoroughly analyzing the estimation error and the communication cost incurred by compression via importance sampling. This required developing new proof techniques (see also the third bullet of our answer to W1). Since analyzing the overall convergence in this setting is too difficult in theory, we resort to extensive empirical comparisons to many baselines, containing the recently proposed M3 method by Gruntkowska et al. '24. The authors prove a reduced communication cost for reaching an $\varepsilon$-stationary point compared to previous schemes such as [2], and moreover also provide theoretical results for the non-convex case. Hence, we only compare to the most recent method of Gruntkowska et al. '24, and show the superiority of our schemes.
> - **W3/Q3**: As mentioned by the Reviewer in S6, we present in the paper extensive experiments for different state-of-the-art datasets, as they appear in the current literature, mainly to enable an easy comparison to alternative proposals. Increasing the complexity of the dataset/model is always of interest, and is possible given the code we supply, but challenging given that we would like to compare our method to multiple existing methods.

---

> > ### Author Response · Authors · 2024-11-20
> >
> > - **W4/Q4**: We agree that maintaining shared randomness has its cost. However, we find the study of the size of shared randomness an aspect of secondary importance only, compared to the communication cost, which is a major bottleneck in FL. While shared randomness could theoretically be limited in size, in practice it is easy to maintain shared randomness with negligible overhead. For example, the standard Tensorflow pseudo-random number generator Philox 4x32-10 has a cycle length of $2^{128} \approx 3.4 \cdot 10^{34}$. Hence, even a single initialization with a common seed should provide sufficient randomness for the entire training process in most cases. Since shared randomness is not privacy critical here, its availability is not of major concern. Hence, we opted to focus the attention of the paper on the analysis of communication cost, and assumed the availability of shared randomness.
> > - **W5/Q1**: We tailor our analysis to the special case of Bernoulli distribution since we believe this specialization preserves tightness and leads to practically relevant theoretical guarantees. Nonetheless, we believe that the novel proof method we present could be modified to various classes of distributions.
> > - **W6/Q7**: For the sake of clarity, in the paper we restrict the analysis to a fixed number of importance samples (nIS), block sizes (B), and choice of priors (p). Our experiments have shown that, while increasing nIS beyond the ones used in our algorithms slightly improves the convergence over the number of epochs, the convergence w.r.t. the communication cost did not significantly improve. The block size is mainly limited by the system resources at hand, and one would choose the largest possible for best efficiency while complying with memory resources. While we commented on the choice of the priors in the paper, we did not extensively elaborate for reasons of brevity. We investigated many different priors, and found that the previous global model serves as a good prior in almost all cases. With high heterogeneity, it might be beneficial to use different convex combinations of the previous global model with the latest posterior estimate of a certain client (see the formulation in the paper, line 265) as a prior, but the gains we observed were minor. Hence, we settled on the former global estimate for simplicity in presenting the algorithm. We added an explaining paragraph to the experimental details in Appendix D, page 19, lines 1005--1015.
> > - **W7/Q5**: Indeed, the sampling mechanism may add compute overhead depending on the implementation and the system resources. Under a tight communication budget, this could be a necessity. We added a short remark at the end of page 19 for clarification.
> > - **Q6**: We consider $10$ clients as a meaningful regime for our schemes. Increasing the number of clients is only expected to positively affect the convergence, since more samples on the uplink are averaged, and since according to our method more samples per client on the DL will be used. Hence, although the diversity of global models seen by the client may increase, their quality will increase, and the average global model estimate will improve. This will ultimately dominate the convergence as long as the clients' estimates are not too dissimilar. We chose $10$ clients as a rather competitive regime (not too small number of clients, not too large to make the comparison unfair).

---

> > > ### Comment · Reviewer_on4o · 2024-11-21
> > > **Follow up discussion**
> > >
> > > Thank you for the detailed answer. However, I still have the following concerns:
> > >
> > > - (W2/Q2): While the authors provide a valid rationale, the absence of convergence analysis remains a significant gap. Without convergence guarantees, the theoretical robustness of the method is questionable.
> > >
> > > - (W6/Q7): More detailed sensitivity analyses regarding hyperparameters would address my concerns about the method's robustness across different settings.
> > >
> > > - (W4/Q4): While the authors provide a plausible explanation, a more empirical assessment or discussion of potential scaling issues (e.g., latency in synchronizing shared randomness across a vast number of clients) would provide a more comprehensive answer.
> > >
> > > - (W5/Q1): The authors suggest that their novel proof methods could be adapted to other distributions, maintaining tightness and practical relevance. While promising, this assertion remains speculative without concrete evidence or examples.
> > >
> > > - (W7/Q5): A more detailed analysis or empirical measurements of the computational and energy costs introduced by their methods would provide valuable insights into the practical feasibility and trade-offs involved, particularly for edge or mobile FL scenarios where resources are constrained.
> > >
> > > - Q6: One of the primary motivations behind the proposed BICOMPFL framework is to reduce communication costs. While demonstrating efficacy with $10$ clients shows promise, it doesn't necessarily translate to effectiveness in environments with significantly more clients, where communication bottlenecks are more pronounced. The comparative performance against baseline methods may vary as the number of clients increases. Some algorithms might scale better, maintaining or even improving their relative performance, while others could degrade. The authors mention that the communication savings achieved by BICOMPFL might be more impactful with an increased number of clients. However, this needs empirical validation.

---

### Official Review · Reviewer_TyDv · 2024-11-01

**Soundness:** 4
**Presentation:** 4
**Contribution:** 4
**Rating:** 6
**Confidence:** 3

**Summary:**

The manuscript considers federated learning under uplink and downlink communication constraints. Several distributed learning strategies based on importance sampling are proposed. The strategies are designed under varying assumptions on the availability of shared common randomness between the server and clients. Theoretical guarantees are provided on the convergence rate of the sampled parameter distribution to that of the target distribution, as a function of the number of samples generated in uplink and downlink operations, which in turn determines the communication rate. Extensive empirical results are provided to compare the performance of the proposed strategies with existing federated learning methods, including those specifically designed for communication-constrained learning scenarios.

**Strengths:**

- The manuscript is well-written. There are few typographical errors if any, and the presentation is easy to follow.
- The topic is of interest to the community, communication constraints, especially in the uplink direction, are a significant roadblock in the implementation of federated learning mechanisms.
- The application of importance sampling, and the several variations considered in the work, is novel in the context of federated learning to my knowledge.
- The theoretical analysis is sound and I have verified the proof steps to extent possible by the limited review time.
- The literature review is comprehensive and covers many of the relevant works.
- The empirical evaluations are comprehensive and provide performance comparisons with a range of existing methods in several different scenarios and benchmark datasets.

**Weaknesses:**

- The computational and storage complexity of the proposed strategies should be evaluated and discussed. A main motivation of the work is the communication-constrained nature of many federate learning scenarios. For instance, in scenarios where clients comprise of mobile edge devices, uplink communication rate may be constrained. However, in such scenarios, the computation and storage complexity of the clients is also limited. As a result, in addition to communication cost, the computation and storage cost need to be considered.
- Note that a main motivation of federated learning is to avoid sharing the client's datasets with the server and with other clients due to privacy considerations. The proposed methods, especially Algorithm 2 which considers global randomness, require that the server share each of the client's updates with other clients. This may not be desirable due to privacy related considerations.
- The experiments consider only 10 clients. It is not clear to the reviewer if some of the proposed methods would scale to larger numbers of clients. Specifically, in Algorithm 3, the clients have different estimates of the global model, since the exact model is not shared with the clients, and it is only estimated in the downlink using important sampling. This may cause the model to not converge when the number of clients is large. Theoretical guarantees or experimental demonstrations of convergence for larger numbers of clients would strengthen the results.

**Questions:**

- Following the first comment in the weaknesses section, please clarify the storage cost and computational cost, compared to existing state-of-the art.
- Following the third comment in the weaknesses section, please comment on the convergence of the model for larger number of clients. Theoretical guarantees or experimental demonstrations of convergence for larger numbers of clients would strengthen the results.
- The empirical results show significant improvements in terms of total communication cost and communication cost per parameter for a fixed number of epochs. However, it is not clear if the prior methods converge at the same pace as the current method. That is, if the previous methods converge faster, then they may not need to be trained for the same number of epochs yielding a better communication cost. An experimental result showing the accuracy vs epoch curve of the proposed methods vs benchmarks would clarify this question.
- The plots combine the uplink and downlink communication costs, however, these would not always be of similar importance in practice. Please comment on or provide experimental results on the UL and DL costs separately.
- In the plots for total communication cost per parameter, it is unclear whether the total is per epoch or over all training epochs.
- Minor Comment: In line 824 on Page 16, the Bernoulli parameter $\frac{(L+1)(p+\eta_\delta)}{n_{IS}}$ could theoretically be greater than one, e.g, by taking $L=n_{IS}-1$, $n_{IS}$ small enoguh, and $\eta_{\delta}$ and $p$ large enough. It appears that there needs to be additional constraints on $n_{IS}$ and $p$ to avoid this.

---

> ### Author Response · Authors · 2024-11-20
>
> - **S1-S6**: We thank the reviewer for the positive evaluation of our work
> - **W1/Q1** Storage and computation costs are indeed relevant in an FL setting. Since we leverage the former global model as a prior, the additional storage cost incurred is limited to storing the estimate of the former global model at each client until the next iteration, which is usually not a bottleneck. This can be even cheaper than some of the baselines, which require storing data for momentum and error-feedback. Importance sampling introduces additional computation overhead for sampling operations, which is usually higher than classical compression schemes. While the computation overhead can be optimized to limit the necessary compute power, some overhead usually remains. We added an explanation at the beginning of page 20 (lines 1033--1038).
> - **W2**: We would like to highlight that our focus here is on the communication efficiency of FL. It is well-known that standard FL schemes, such as FedAvg, do not provide any formal privacy guarantees, and need to employ additional privacy mechanisms such as randomization or secure multi-party computation. These techniques can also be applied in our framework to provide formal privacy guarantees. Moreover, the additional privacy leakage of Algorithm 2 compared to standard FL approaches is mild as long as the federator does not share the clients' identity with the corresponding indices. Indeed, when simply sharing the union of the indices, no individual information is leaked beyond an estimate of the global model, similar to standard FL approaches.
> - **W3/Q2**: Increasing the number of clients is expected to positively affect the convergence since more samples would be averaged on the UL, and according to our method, more samples per client will be used on the DL. Hence, although the diversity of global models seen by the clients may increase, their quality will also improve, and the average global model estimate will converge faster, which will ultimately dominate the convergence as long as the clients' estimates are not too dissimilar. We chose to run the simulations for $10$ clients as a rather challenging regime for our schemes.
> - **Q3**: We now provide additional plots showing the accuracy of our proposed schemes over the number of epochs. We included such plots in Fig. 11 at the end of Appendix E, and conclude that our proposed method is comparable to the baselines in terms of accuracy versus epochs. Nevertheless, we would like to emphasize that our earlier plots show the accuracy reached for all the schemes versus the total number of communicated bits, which, when communication is the bottleneck, would be the stopping criterion of interest.
> - **Q4** While we show the total communication cost in the plots, we provided the UL and DL bit rates separately in the Tables in the Appendix. Additionally, we now provide the total bit rates of our scheme compared to the baselines when a broadcast link is available from the federator to the clients, i.e., the weight of the DL cost is reduced. Those comparisons are added to all the experimental results in the Appendix. Even in such regimes, which we think is one of the most important examples in which UL and DL are not equally important, our schemes, especially the ones with global shared randomness, still exhibit good performance.
> - **Q5**: We mean the cost per parameter per epoch. We clarified this in the manuscript (line 352 on page 7).
> - **Q6**: We apologize for this confusion. $\rho + \eta_\delta$ is always bounded by $1$ from above. Otherwise, the concentration (line 778) could be trivially bounded. We added a respective footnote to page 15 for clarification.

---

> > ### Comment · Reviewer_TyDv · 2024-11-24
> >
> > Thank you to the authors for their detailed response. Most of my questions have been addressed. My only remaining concern is how the proposed methods would scale to a larger number of clients. While the authors explain that simulations were conducted with 10 clients as it represents a more challenging scenario, it would have been helpful to see experiments involving a larger client base to evaluate the scalability and resulting performance. That said, I understand that such experiments may not be feasible within the limited rebuttal period. I have no further questions and will maintain my score.

---

### Official Review · Reviewer_sMk3 · 2024-11-03

**Soundness:** 2
**Presentation:** 1
**Contribution:** 2
**Rating:** 3
**Confidence:** 5

**Summary:**

The paper discusses bi-directional compression schemes in a federated learning framework assuming global and private common randomness in the model.

**Strengths:**

The attempt to study the shared randomness in a FL setting seems interesting.

**Weaknesses:**

The are some issues with the paper:

- An important issue is that, given this paper considers various schemes within a system model that includes shared common randomness, the performance of previous algorithms like FedAvg should also be re-evaluated under the assumption of shared randomness, with results reported in the paper for comparison. Specifically, it is unclear whether the improvement shown in Fig. 2 arises from the availability of shared randomness or from the algorithm itself. If the improvement is primarily due to the former, the comparison may not be entirely fair.

- Another important issue with the paper is that the results presented in Fig. 2 and Theorem 1 should account for the size of the common randomness. There is no discussion on whether this size is sufficiently large or if specific assumptions are being made about it.

- Section 3 primarily focuses on the mathematical details of the algorithm, with limited emphasis on the intuition behind it. To improve clarity, these mathematical details should be moved to the appendix, allowing this section to concentrate on explaining the intuition and comparing the algorithm with previous approaches.

**Questions:**

Can the authors refine the previous schemes' performances assuming the availability of shared randomness in the system model?

---

> ### Author Response · Authors · 2024-11-17
>
> - **S1**: We agree that shared randomness in FL is quite interesting, but we would like to emphasize that our high level question is even more interesting, and it is how bi-directional compression can be achieved in stochastic FL. Our method is based on a  principled compression method through importance sampling under side information, and this indeed brings the aspects of shared randomness into the picture. In other words, while the impact of shared randomness (or its unavailability) is interesting, our focus is on the communication cost of FL, as it is a major bottleneck.
> - **W1**: Our focus is not the investigation of a particular training process, but we rather aim  to demonstrate that stochastic FL allows for alternative compression schemes, which can leverage side-information in a principled way (in the form of the prior global model), and allow the exact quantification of the communication cost. We would like to highlight that non-stochastic FL methods, such as FedAvg, cannot benefit from shared randomness. Hence, the improvements of our scheme are not a direct result of the availability of shared randomness, but primarily from the stochastic formulation of the compression method.
> - **W2**: We find the study of the size of shared randomness an aspect of  secondary importance, compared to the  communication cost, which, as said, is a major bottleneck in FL. While shared randomness could theoretically be limited in size, in practice it is easy to maintain shared randomness with negligible overhead. For example, the standard Tensorflow pseudo-random number generator Philox 4x32-10 has a cycle length of $2^{128} \approx 3.4 \cdot 10^{34}$. Hence, even a single initialization with a common seed should provide sufficient randomness for the entire training process in most cases. Since shared randomness is not privacy critical here, its availability is not a major concern. Hence, we opted to focus our attention on the analysis of communication cost, and assumed the availability of sufficient shared randomness.
> - **W3**: We agree that complicated mathematical details should be deferred to later sections or the appendix, and the focus in Section III should be on intuitive description of the algorithm. Following your comment,  we are happy to move the part between lines 264 and 292 to the Appendix, and use the freed-up space for an intuitive explanation of the conceptual differences of our compression method from the baseline schemes.
> - **Q1**: As detailed in the response to W1, the baseline compression methods are deterministic; and thus, cannot benefit from the availability of shared randomness. Their approach to compression is substantially different from ours, which is one of the core novelties of our work: rather than sending exact model updates, we enable the receivers to recover random updates by sampling from the correct distribution, which has a significantly lower communication requirement. As mentioned in our response to W2, shared randomness is typically available with negligible cost.

---

> > ### Comment · Reviewer_sMk3 · 2024-11-22
> > **missing key-length analysis and comparison with prior works**
> >
> > Thanks to the authors for response. I am still not convinced with it.
> >
> > First, regarding the key size, if it is indeed infinite, this should be explicitly stated in the paper. Currently, such a clarification is missing. Moreover, it is important to evaluate whether an infinite key length leads to a meaningful scenario. While the practicality of this scenario could potentially be justified, the critical point is to examine how different evaluation metrics vary with respect to key length. Addressing this question requires further research.
> >
> > Second, regarding the comparison with prior algorithms: if the system model in previous works is deterministic while the current setting is stochastic, the comparison does not seem entirely fair since the two settings are fundamentally different. A fair comparison would require adapting the previous algorithms to the stochastic setting.

---

### Official Review · Reviewer_a6x3 · 2024-11-04

**Soundness:** 4
**Presentation:** 4
**Contribution:** 3
**Rating:** 5
**Confidence:** 4

**Summary:**

This paper proposes2 algorithms for bi-directional version of the work of ISIK et. al.  The first algorithm assumes that globally shared randomness is available, and the second is for the case when only a private random key between each client and the central federator is available. The authors provide experiments showing improvements in the communication bandwidth for the same performance. Some analysis is also provided.

**Strengths:**

The improvements in the communication bandwidth, for the same performance and theoretical analysis.

**Weaknesses:**

Modification of Isik et. al which by itself is more of a narrow problem in Federated learning. There exists lots of proposals in this direction, so the paper while novel is not highly original. For instance, the paper by Philippenko et. al, Avdiukhin et. al, etc. This paper results are good, but I am not sure if it is good enough for a competitive venue like ICLR.

**Questions:**

What is the justification for |q_j - p_j| being uniformly bounded? It does not seem like a mild assumption to me.

---

> ### Author Response · Authors · 2024-11-17
>
> - **S1**: We thank the Reviewer for the positive evaluation of our work
> - **W1**: As the reviewer indicated, there  are many proposals in this direction. However, all of them, to the best of our knowledge, focus on non-stochastic FL. The paper by Avdviukhin et al. focuses on one specific aspect of FL with bi-directional compression; it considers an asynchronous scenario, where model updates in the form of gradients are accumulated until reaching a certain norm. While such ideas can be incorporated into our framework, our approach differs conceptually due to the focus on stochastic FL. Similarly, Philippenko et al. and Gruntkowska et al. focus on the specific aspect of sending different compressed updates to each client in the context of non-stochastic FL. While their methodology was an inspiration to our work, we argue that none of these works is tailored to stochastic FL, which provides an advantageous alternative to classical approaches.
>
>    Our intention is not to extend FedPM to include downlink (DL) compression, but rather to address DL for general stochastic FL. This is well-motivated since  many works (cf. review in Sec. 6) study the specific caveats of joint uplink (UL) and DL compression in non-stochastic FL, requiring refined analysis and methods. To the best of our knowledge, we are the first to bring this aspect to stochastic FL. There are several  challenges associated with the inclusion of DL compression in the model:
>     1. The role of shared randomness: In UL-only compression, only private shared randomness is required. However, in bi-directional compression, the absence of global shared randomness may lead to divergence of the learning process; in particular when downlink compression is not carefully designed, and clients start their local training from very different models. This aspect is non-critical for deterministic compression methods; and therefore, was not previously explored. However, this aspect is crucial for stochastic FL via importance sampling. Carefully examining the role of different notions of shared randomness is a novel aspect of our paper.
>     2. Analysis and choice of side-information in the case of synchronized and non-synchronized compression, that is, a  distinction between  global shared randomness and private shared randomness. The introduction of additional noise in the training process by individual compression noise on each of the clients strongly affects the learning process and depends crucially on the prior choice. Controlling the noise introduced on the DL as a function of the choice of the prior is thus highly critical. The choice of priors we propose in the paper may seem natural, but is in fact the result of a careful investigation of different priors in different settings of heterogeneity (we only touch upon this in the paper). Conditioned on the prior choice, we analyze the error made by the importance sampling-based compression.
>     3. Analysis of communication cost: We develop a new proof strategy to provide analytical results for the quality of the federator's estimate of the client updates. By contrast, Isik '24 just used generic results from importance sampling, which are inapplicable to the case of bi-directional compression. This result can also be seen as a refined version of the result of Isik '24. Specializing the analysis to Bernoulli parameters enables a tractable  analysis that preserves tightness, which we find more practically relevant than existing theoretical bounds. We believe the proof method is novel, and can be of interest on its own as it can be used for deriving similar results for other types of distributions.
>
>     In general, we would like to highlight that our method applies to any stochastic FL framework. We have adopted FedPM of Isik et al. only for experimental validation. This is motivated by the fact that it achieves high accuracy, but also allows to accurately quantify the communication cost. Given your comment, we understand that the submitted manuscript did not clearly convey the general challenges of bi-directional compression for stochastic FL. We have added a few clarifying sentences in the updated version of the manuscript based on the above discussion, and we believe that this is now clear.
> - **Q1**: The bound on $|q_j - p_j|$ arises naturally from the properties of the learning algorithm. In particular, the training process effectively employs point-wise optimization with respect to a KL-proximity constraint, which immediately limits how far a client's training parameter $q_j$ will diverge from $p_j$. In case other optimization algorithms are used, such a constraint can be ensured by a projection step. We did test such projections in experiments and found them to be effective; however, we only briefly mentioned this due to space limitations as this is not the focus of the paper.

---

> > ### Comment · Reviewer_a6x3 · 2024-11-22
> > **Thanks for your response**
> >
> > Thanks for your response.  I understand the differences between your paper and those in the literature.  But I am not convinced (as many other reviewers are not) that these differences are substantial enough to warrant publications in the ICLR.
> >
> > I was already generous with my score, and I am keeping it.

---

### Official Review · Reviewer_FzVZ · 2024-11-05

**Soundness:** 3
**Presentation:** 3
**Contribution:** 2
**Rating:** 5
**Confidence:** 3

**Summary:**

This work is about communication-efficient FL that leverages side information to transmit samples from the desired distribution through importance sampling. Unlike in most FL works, this paper assumes that downlink communication is also bandwidth-constrained. Convergence analysis is given and experimental results on accuracy versus total bits required validate the method.

**Strengths:**

The results are strong and practical merits are high.

**Weaknesses:**

In a nutshell, the proposed idea is to bring in a bidirectional element to FedPM of Isik et al. 2023. Probabilistic mask training, synchronized randomness and importance sampling are all from the existing work of Isik et al. 2023 and 2024. The only significant new element seems to be the introduction of downlink cost (as described by the n_DL dependent term in the first equation of section 5) in the analysis and performance evaluation. Thus, overall novelty is not high.

Also, the authors would be overestimating the downlink communication cost in wireless scenarios since downlink is achieved by broadcasting and the effective bandwidth requirement should be reduced by the factor n. In this case, the effect of including downlink cost would be marginal, questioning the relative merits of the proposed approach.

**Questions:**

About the total communication cost in bits per parameter in Fig. 1, how is the uplink/downlink cost combined? As said above, does the overall cost reflect the difference in the bandwidth requirement for broadcasting vs separate one-to-one links?

Maximum accuracy is higher with the compressed schemes than FedAvg in Fig. 2? How is this so?

I would like to hear authors' response to novelty issue raised above.

---

> ### Author Response · Authors · 2024-11-17
>
> - **S1**: We thank the Reviewer for their positive evaluation.
> - **W1**: Our intention is not to extend FedPM to include downlink (DL) compression, but rather to address DL for general stochastic FL. This is well-motivated since  many works (cf. review in Sec. 6) study the specific caveats of joint uplink (UL) and DL compression in non-stochastic FL, requiring refined analysis and methods. To the best of our knowledge, we are the first to bring this aspect to stochastic FL. There are several  challenges associated with the inclusion of DL compression in the model:
>     1. The role of shared randomness: In UL-only compression, only private shared randomness is required. However, in bi-directional compression, the absence of global shared randomness may lead to divergence of the learning process; in particular when downlink compression is not carefully designed, and clients start their local training from very different models. This aspect is non-critical for deterministic compression methods; and therefore, was not previously explored. However, this aspect is crucial for stochastic FL via importance sampling. Carefully examining the role of different notions of shared randomness is a novel aspect of our paper.
>     2. Analysis and choice of side-information in the case of synchronized and non-synchronized compression, that is, a  distinction between  global shared randomness and private shared randomness. The introduction of additional noise in the training process by individual compression noise on each of the clients strongly affects the learning process and depends crucially on the prior choice. Controlling the noise introduced on the DL as a function of the choice of the prior is thus highly critical. The choice of priors we propose in the paper may seem natural, but is in fact the result of a careful investigation of different priors in different settings of heterogeneity (we only touch upon this in the paper). Conditioned on the prior choice, we analyze the error made by the importance sampling-based compression.
>     3. Analysis of communication cost: We develop a new proof strategy to provide analytical results for the quality of the federator's estimate of the client updates. By contrast, Isik '24 just used generic results from importance sampling, which are inapplicable to the case of bi-directional compression. This result can also be seen as a refined version of the result of Isik '24. Specializing the analysis to Bernoulli parameters enables a tractable  analysis that preserves tightness, which we find more practically relevant than existing theoretical bounds. We believe the proof method is novel, and can be of interest on its own as it can be used for deriving similar results for other types of distributions.
>
>     In general, we would like to highlight that our method applies to any stochastic FL framework. We have adopted FedPM of Isik et al. only for experimental validation. This is motivated by the fact that it achieves high accuracy, but also allows to accurately quantify the communication cost. Given your comment, we understand that the submitted manuscript did not clearly convey the general challenges of bi-directional compression for stochastic FL. We have added a few clarifying sentences in the updated version of the manuscript based on the above discussion, and we believe that this is now clear.

---

> > ### Comment · Reviewer_FzVZ · 2024-11-20
> >
> > Thanks for further comments and explanations. Relative to the FedPM-KLM idea presented in Isik et al. 2024, however, I am still not convinced that  there are significant new concepts in Algorithms 1, 2 and 3 proposed here. For example, I see the compression in the downlink in Algorithm 2, but once a DL bandwidth constraint is imposed, broadcasting the uploaded sample indices from all other clients (either this or broadcasting indices of new samples that best represent the updated global model) is a fairly natural thing anyone would do. Likewise, the DL element in Algorithm 3 is conceptually based on FedPM-KLM - namely, importance sampling and use of priors based on probability mask. The choice of the prior you make in DL  is also consistent with the spirit of  FedPM-KLM and is not surprising. The introduction of private shared randomness seems new, but it is not clear how practically meaningful this setting is.

---

> > > ### Author Response · Authors · 2024-11-20
> > >
> > > We appreciate your engagement in a discussion with us. Let us emphasize the key contributions and relevance of our paper to the ICLR community. Bi-directional non-stochastic FL methods obtain stable results only when embellished with various upgrades such as momentum. By contrast, even a vanilla version of our method significantly outperforms their performance. Additional results in the updated paper show that this is true even if a broadcast link is available. We could have increased the perceived novelty of our method by complicating it, e.g., using different priors depending on the heterogeneity of the clients’ data. However, we opted not to do so, because these just lead to minor improvements based on our experiments, despite introducing substantial complexity. We would also like to bring into attention the theoretical analysis, which goes far beyond known theory. This is an additional main novel contribution. We hope that the paper would be judged based on these achievements.

---

> > > > ### Comment · Reviewer_FzVZ · 2024-11-26
> > > >
> > > > Thanks for your efforts. But I am not convinced. I maintain that this work closely follows FedPM-KLMS.

---

> ### Author Response · Authors · 2024-11-17
>
> - **W2**: We measure the  overall bitrate by the sum of UL and DL bitrates, i.e., in the absence of broadcast links. There are two reasons for that. First, broadcast links are not always available in practice, e.g., in the common wireless scenario of clients that are spread geographically. Second, while the presence of broadcast links will make the downlink cost marginal, this will only increase our gain compared to other bi-directional methods, which we thought might look like an unfair comparison.  Indeed, let's assume that broadcast links are available. There are two options. In the case of global randomness, we would scale down the bitrate per parameter and per client by the number of clients, reaching the benefits of UL-only compression, which are even better than the overall bitrates for the other baselines. In the case of private randomness, our method cannot benefit from broadcasting, so we scale down the bitrates of the baseline schemes but keep the suboptimal bitrate of our scheme for fairness. Even in this case, our approach will still be significantly better due to savings on the UL. One can still argue that resorting to no-compression in the DL is a viable option. However, for moderate number of clients (say $10-100$), even with the availability of broadcasting, no compression would lead to $~0.32-3.2$ bits/p.p. on the DL, significantly higher than the UL cost, hence drastically worsening the overall bitrate. Compression on the DL would only be obsolete in the regime where $n$ is very large (say significantly larger than $1000$), which is a generic argument that holds for every bi-directional compression scheme (that is, all the baselines we compare with). We added to all the experiments and schemes the bitrates achievable when a broadcast link is available from the federator to the clients. The added columns and the textual explanation are highlighted in blue for convenience. To conclude, the DL cost is typically not marginal, and our method has merits even when it is.
>
> - **Q1**: As explained in the answer to W2, we sum the cost of uplink and downlink communication per parameter, i.e., equal scaling. Following the reviewer's implicit request, we will add to each of the tables a column that compares the bitrates in the presence of broadcast links.
> - **Q2**: In this particular instance, FedAvg is negligibly worse (please note the y-axis) on average than DoubleSqueeze; its accuracy lies  well within the standard deviations reported in Tab. 7. Hence, this deviation is just a result of a statistical fluctuation.

---

### Meta-Review · Area_Chair_Bh9j · 2024-12-17

**Metareview:**

This paper considers leveraging side information in the form of prior information in stochastic federated learning (FL). The motivation is that the addition of this side information can reduce the uplink communication cost. There were some remaining concerns about the paper even after the discussion phase. I agree with these concerns, and hence assess that the paper does not cross the bar for publication at ICLR.

Firstly, the proposed method is incremental compared to Isik et al. (2023). It only introduces a bidirectional element to Isik et al. (2023). The only addition is the inclusion of the downlink cost. Two reviewers (FzVz and a6x3) mention this, and feel that this modification is not sufficiently novel.

Secondly, Reviewer TyDv feels that the work would have benefited from a more comprehensive empirical evaluation—particularly to address concerns regarding scalability of the proposed approach—and a more complete theoretical evaluation—particularly derivation of convergence guarantees.

**Additional Comments On Reviewer Discussion:**

The authors tried to highlight the differences between their work and those in the literature. However, the reviewers remain unconvinced about the novelty, especially w.r.t. FedPM-KLMS of Isik et al. (2023).

---

### Decision · Program_Chairs · 2025-01-22

Reject